# Plant Molecular Pharming and Plant-Derived Compounds towards Generation of Vaccines and Therapeutics against Coronaviruses

**DOI:** 10.3390/vaccines10111805

**Published:** 2022-10-26

**Authors:** Srividhya Venkataraman

**Affiliations:** Virology Laboratory, Department of Cell & Systems Biology, University of Toronto, Toronto, ON M5S 3B2, Canada; byokem@hotmail.com

**Keywords:** coronavirus, SARS-CoV, MERS-CoV, SARS-CoV-2, molecular pharming, plant-based vaccines, therapeutics, monoclonal antibodies, phytochemicals, regulatory issues

## Abstract

The current century has witnessed infections of pandemic proportions caused by Coronaviruses (CoV) including severe acute respiratory syndrome-related CoV (SARS-CoV), Middle East respiratory syndrome-related CoV (MERS-CoV) and the recently identified SARS-CoV2. Significantly, the SARS-CoV2 outbreak, declared a pandemic in early 2020, has wreaked devastation and imposed intense pressure on medical establishments world-wide in a short time period by spreading at a rapid pace, resulting in high morbidity and mortality. Therefore, there is a compelling need to combat and contain the CoV infections. The current review addresses the unique features of the molecular virology of major Coronaviruses that may be tractable towards antiviral targeting and design of novel preventative and therapeutic intervention strategies. Plant-derived vaccines, in particular oral vaccines, afford safer, effectual and low-cost avenues to develop antivirals and fast response vaccines, requiring minimal infrastructure and trained personnel for vaccine administration in developing countries. This review article discusses recent developments in the generation of plant-based vaccines, therapeutic/drug molecules, monoclonal antibodies and phytochemicals to preclude and combat infections caused by SARS-CoV, MERS-CoV and SARS-CoV-2 viruses. Efficacious plant-derived antivirals could contribute significantly to combating emerging and re-emerging pathogenic CoV infections and help stem the tide of any future pandemics.

## 1. Introduction

During the 21st century, the world has witnessed infections by emerging Coronaviruses such as the severe acute respiratory syndrome coronavirus (SARS-CoV) in 2003, Middle East respiratory syndrome coronavirus (MERS-CoV) in 2012 and the Covid19 virus (SARS-CoV-2) in 2019, which spread rapidly, causing life-endangering respiratory infections associated with high morbidity and mortality [1,2,3,4]. While the SARS-CoV virus has disappeared, the other two Coronaviruses are still actively infecting the human population; particularly, the SARS-CoV-2 virus is rampantly disseminating across the world and poses great challenges to human health world-wide. 

The SARS-CoV-2 or the Covid19 virus has been proclaimed as a pandemic on account of its ability to spread rapidly between humans within a short duration. These unprecedented events reveal that novel Coronaviruses or their variants could emerge in the imminent future [5]. Such frequently occurring virus infection outbreaks have led to several global crises with tremendous negative consequences on human health and hence the global economy, impacting financial stability and human lifestyles [6]. Therefore, there is a compelling need to develop therapeutic and prophylactic interventions such as antibodies, vaccines, therapeutic proteins and diagnostic reagents against emerging coronavirus infections to stem the ongoing pandemic and preclude any future outbreaks [7,8,9].

Presently, several vaccines and therapeutic proteins have been approved against Covid19 with many of them under clinical trials. Testing of vaccines is time-consuming, requiring large amounts of drug products to achieve clinical potency [10]. In this context, biopharming in plants affords several advantages such as safety, low-cost and ease of scalability towards the expression of recombinant biopharmaceuticals when compared with conventional mammalian and bacterial expression platforms [11,12]. 

Plants circumvent several challenges involved in the production of biopharmaceuticals, particularly in pandemic situations, by functioning as rapid, efficacious schemes for manufacturing in bulk amounts on large scales, thus meeting with the requirements of biopharmaceuticals to ameliorate human diseases world-wide. Moreover, previous accmplishments in the successful production of recombinant proteins such as monoclonal antibodies and vaccines against Influenza, Ebola and HIV diseases has shown that plants can function as appealing platforms for generating therapeutic proteins and other biologics to mitigate disease outbreaks [13]. 

The current review highlights emerging coronavirus infections, the similarities between various beta-Coronaviruses of important concern to human health, and their mitigation using plant-derived immuno-therapeutics and vaccines, in addition to descriptions of plant-based natural metabolites and phytochemicals of great applicability in the prevention/treatment of coronavirus diseases. 

## 2. An Overview of Coronavirus Classification, Molecular Biology and Genome Organization

Coronaviruses (CoVs) are important enveloped viruses composed of single-stranded, positive-sense RNA genomes that cause a wide spectrum of respiratory, neurological, gastrointestinal and renal disorders in humans, other mammals and birds [14]. CoVs are members of the order Nidovirales within the family Coronaviridae containing four genera inclusive of the Alphacoronavirus and the Betacoronavirus primarily infecting mammals, as well as the Gammacoronavirus and the Deltacoronavirus which cause infections in birds [15,16]. These viruses characteristically appear crown-like under the electron microscope, hence the name ‘Coronaviruses’ [17]. The CoV genomes range in size between 27.3 and 31.3 kb containing 6 to 11 operative open reading frames that code for structural and non-structural polypeptides as well as other accessory proteins. 

Seven coronavirus types have so far been known to infect humans. Human Coronaviruses HCoV-HKU1, HCoV-229E, HCoV-OC43 and HCoV-NL63 generally induce mild clinical symptoms in addition to self-limiting infection of the respiratory tract; however, in immune-compromised patients, they may evoke severe symptoms [18,19,20]. The recently reported SARS-CoV, SARS-CoV-2 and MERS-CoV are highly pathogenic and present serious threats to human health. 

The classification, epidemiology, genome organization, molecular characteristics, transmission, symptoms, vaccines and diagnostic methods for all three highly infectious Coronaviruses are listed in Table 1. 

Both SARS-CoV and Covid19 viruses share similarity in using the angiotensin-converting enzyme 2 (ACE2) receptor to initiate infection [7]. The SARS-CoV begins infecting target cells by binding to the cellular receptor via the receptor binding domain (RBD) occurring in its spike protein S1 subunit leading to pre-fusion and entry of the virus. The ACE2 receptor is a major proinflammatory molecule that is expressed in cells, especially in the lungs, liver, kidneys and intestines [21,22]. The RBD of SARS-CoV2 interacts with the ACE2 receptor at higher efficiency rates about 10–20 fold greater than that of the SARS-CoV RBD with the ACE2 receptor [23]. 

On the other hand, the dipeptidyl peptidase 4 (DPP4) also called CD26, occurs as the MERS-CoV cellular receptor [7]. The RBD of MERS-CoV S protein binds to the DPP4 receptor on the surface of the cells leading to S protein cleavage into S1 and S2 subunits enabled by host proteases followed by the fusion of the cellular and virus membrane, thus releasing the virus genome into the host cell [24,25]. Besides humans, bats, non-human primates and camels function as hosts for MERS-CoV. However, ferrets, mice and hamsters are not susceptible to MERS-CoV infection despite having the DPP4 cellular receptor [26]. DPP4 occurs as a type-II transmembrane dimeric molecule on the surface of epithelial cells present in the lungs, liver, prostate and small intestine [27]. 

## 3. Genetic Similarities between MERS-CoV, SARS-CoV and SARS-CoV-2

Between the various CoV subtypes, beta CoVs, are responsible for serious and fatal diseases while alpha CoV induce mild infections. The genomic sequences of the SARS-CoV, SARS-CoV-2 and MERS-CoV are largely similar; however, SARS-CoV-2 shows major differences in the composition of its genome compared to that of its predecessors, MERS-CoV and SARS-CoV [28,29]. Genomic analysis indicates that SARS-CoV has close relationship with bat CoV (96%) and pangolin CoV (86–92%), which suggests that bats could be the primary reservoir of SARS-CoV-2 [19,30,31]. 

MERS-CoV bears close relationship with two bat CoVs ((HKU4, HKU5) and has been proposed to originate from bats with dromedary camels functioning as intermediate host, as discerned from serological investigations [32,33]. MERS-CoV RNA was detected in swabs collected from dromedary camels from Qatar which shared relationship with two of the MERS-CoV human cases [34]. A thorough evolutionary correlation study revealed that the MERS-CoV originated from bats by the incidence of recombination events within the S and ORF1ab genes [35,36]. Similar recombination events were also detected in SARS-CoV as regions of likely recombination events were identified using computational genomics [37].

MERS-CoV strains obtained from camels and humans reportedly share more than 99% identity with differences located in S, ORF3 and ORF4b genes [38]. SARS-CoV-2 displays 51% similarity with MERS-CoV and 80% similarity with SARS-CoV [39]. The majority of the coding regions of SARS-CoV-2 contain genomic architecture similar to that of SARS-CoV and CoVs originating from bats. The predicted 12 coding regions are: E, M, lab, 3, 7, 8, 9, 10B, S, N, 13 and 14. The proteins coded for by all three CoVs are for the most part similar in length [40]. Nevertheless, there is major difference in the SARS-CoV-2 S protein length which is greater when compared with the S proteins encoded by MERS-CoV, SARS-CoV and bat CoVs [41]. 

Several similarities occur in terms of pathogenicity and architecture between the SARS-CoV-2 and the SARS-CoV in contrast to that of MERS-CoV. Decision-tree experiments based on mathematical modeling have revealed notable characteristics of the amino acid sequence of the SARS-CoV-2, which is distinct from that of MERS-CoV. Coronaviruses possess similar S proteins that bind to their respective host cells and, additionally, they use the same host protease for activating their S proteins [42]. 

The SARS-CoV-2 S protein has 77% sequence similarity with that of the SARS-CoV, the structural proteins show 90% similarity to that of SARS-CoV and 32.79% similarity to that of MERS-CoV. The S2 receptor-binding domain of SARS-CoV-2 displays 74% similarity to that of SARS-CoV [43]. The SARS-CoV-2 E protein shows 96% and 36% similarity to that of SARS-CoV and MERS-CoV, respectively. while the M protein is 89.59% and 39.27% similar to that of the SARS-CoV and MERS-CoV, and the N protein of SARS-CoV-2 shows 85.41% and 48.47% sequence similarity with that of SARS-CoV and MERS-CoV, respectively [44]. Accessory proteins function as important factors during viral replication and pathogenesis [45,46]. The SARS-CoV-2 RdRp (nsp12) and 3CLpro (nsp5) proteins act as major mediators in virus replication and production of new virions and share high levels of sequence identity with those of SARS-CoV and MERS-CoV [47]. 

Recent investigations have shown that the SARS-CoV ORF3a and ORF8b proteins catalyze proinflammatory cytokine induction, thereby playing a major role in the regulation of macrophage chemotaxis [48,49]. The SARS-CoV and MERS-CoV ORF8b gene products are also involved with the suppression of interferon (IFN-I) induction [50,51]. Yet another investigation reported that the SARS-CoV-2 variant ORF8b interacts with major histocompatibility complex (MHC) and controls its degradation in in vitro cell cultures, implicating ORF8 in immune evasion. Nevertheless, the ORF8 of SARS-CoV-2 shows low level of homology to that of SARS-CoV [52]. In general, there are no homologous accessory proteins found in Coronavirus genera. The ORF3a, 6, 7a, 7b and 9b proteins of SARS-CoV-2 and SARS-CoV share over 80% sequence similarities. 

## 4. Immunological and Clinical Aspects of Coronavirus Infections

### 4.1. MERS-CoV

MERS is presently a typical human coronavirus infection. The MERS-CoV is less transmissible than the other two major CoVs but induces adverse symptoms that lead to higher fatality rates [22]. MERS patients exhibit milder symptoms at the initial stages of infection leading to the later development of dyspnea and further complications causing respiratory failure, with majority (63.4%) of the patients progressing into lethal pneumonia [53]. Deterioration of organ function ensues and then leads to death within 2 weeks following infection [54]. 

Important comorbidities involving mortality in MERS-CoV infections are renal failure and diabetes that lead to poor outcomes. Unlike the abortive infection mechanism of the SARS-CoV, MERS-CoV multiplies in dendritic cells, macrophages and lymphocytes [55,56,57,58]. Virus particles, viral genomes and nucleoprotein expression are found in MERS-CoV-infected cells. Virus multiplication in dendritic cells and macrophages show that host cells act as viral reservoirs that confront immune recognition of the virus by host cells [57]. MERS-CoV induces more transcriptomic changes compared to those caused by SARS-CoV [41]. 

Cells harboring the virus enable systemic spread of the infection into the lymph nodes. Naïve T cells bind to the virus and elicit adaptive immune reactions leading to the release of immense amounts of chemokines and cytokines. The widespread activation channels triggering cytokine production in MERS-CoV infection result in distinct profiles of cytokines in comparison with that of the SARS-CoV infection [41,59]. The reason for the prolific replication in MERS-CoV infection is the greater number of the DDP4 receptors expressed on monocytes and dendritic cells in contrast to the lower ACE2 receptor expression levels which leads to differential disease outcomes. MERS-CoV is capable of infecting cells from various human cell lines as seen in ex-vivo studies [60,61]. 

The DDP4 receptors have been identified in epithelial and endothelial cells found in kidney, liver, intestines and prostate [62,63]. In MERS-CoV infections, the virus spreads throughout the body resulting in high prevalence of systemic events such as septic shock and multi-organ failure. Yet another significant immunopathological feature of the MERS-CoV infection is the antibody-dependent enhancement (ADE), the underlying mechanism of which is associated with augmented membrane fusion. The binding of the S protein RBD and the antibodies lead to increased susceptibility to proteolysis causing conformational alterations in target cells [57]. The binding antibodies increase viral entry through recognizable receptor-based pathways. 

### 4.2. SARS-CoV

Downregulated levels of CD4^+^ and CD8^+^ T lymphocytes, B lymphocytes, natural killer cells and dendritic cell subsets are detectable in SARS patients at the initial stages of infection prior to the use of steroids [64,65]. In the initial week after infection, the SARS illness is presented as an abortive or self-limiting infection of the peripheral blood mononuclear cells, as observed by the occurrence of minus-strand viral RNA that is the replicative intermediate of SARS virus [66]. High levels of monocyte chemotactic protein 1 (MCP-1 [CCL2]), interleukin-8 (IL-8) and plasma chemokines, gamma interferon (IFN-γ)-inducible protein 10 (IP10 [CXCL10]) were consistently observed. Additionally, elevated levels of inflammatory cytokines IL-1β and IL-6 and Th1-related cytokines IFN-γ and IL-12 were detected potentiating an intense inflammatory response [67,68,69,70,71,72].

In one investigation, patients having severe disease showed augmented plasma levels of CXCL10, IFN-α, IFN-γ, and reduced levels of tumor necrosis factor alpha (TNF-α), IL-12p70 and IL-2 during the acute phase of infection. On the other hand, in the late phase these patients had highly enhanced plasma chemokine levels of CCL2, CXCL10 and IL-8, but diminished cytokine levels of TNF-α, IFN-γ, IL-12p70 and IL-2 [73]. Such host responses may be responsible for the assignment and accretion of polymorphs and alveolar macrophages and the elicitation of Th-1 cell-mediated immune response by triggering of natural killer cells and cytotoxic T lymphocytes, respectively. As the SARS-CoV evades the stimulation of IFN-α and IFN-β in vitro in human macrophages [74,75], the absence of antiviral innate immune reactions could allow the unbridled replication of the virus, leading to progressive augmentation of viral load accompanied by systemic proinflammatory response. This condition extends into the second week of infection until the emergence of adaptive immune response, which leads to control of virus replication. Marked upregulation of genes involved in inflammation, apoptosis, pro-coagulation and stress response were observed in the early phase of SARS infection in human liver cancer cell line (Huh7) [76]. This could account for the clinical severity of the SARS infection in relation to the augmented viral load at up to 2 weeks from the onset of illness and the acute inflammatory response as observed from histopathology and serum cytokine profiles. 

Notably, a majority of patients infected with SARS resolved the proinflammatory chemokine and cytokine responses at the severe phase and expressed genes involved in adaptive immunity [77]. On the other hand, susceptible patients exhibited deviated immunoglobulin and IFN-stimulated gene expression levels, unrelenting chemokine levels and deficient antibody production against the SARS spike protein. It was hypothesized that the unregulated IFN responses in the acute phase could lead to impairment of the switch from innate to adaptive immunity. Indeed, patients who recovered showed higher and persistent levels of N-specific and S-specific neutralizing antibody reactions, while susceptible patients showed an initial increase and then a fall in the levels of the antibodies just prior to death, indicating that antibody response could play a major role in determining the final outcome of the SARS-CoV disease [78]. 

### 4.3. SARS-CoV-2

SARS-CoV-2 infects both type II alveolar cells and alveolar macrophages by interaction with the ACE2 receptors on the host cell membrane [79]. Prior to entry into host cells, the SARS-CoV-2 spike protein 1 (S1) undergoes pre-activation by the host furan to expose the receptor-binding domain (RBD) of S1. The RBD binds with strong affinity to the ACE2 receptor thus facilitating effective viral entry. Following the RBD-ACE2 binding, the type 2 transmembrane protease (TMPRSS2) proteolytically cleaves the S1-S2 protein leading to drastic changes in its structure, further exposing the fusion peptide of S2, thus facilitating the fusion of the virus to the host cell [80]. 

Immediately following entry into the cell, the lysosomal cathepsins along with furin activate the S protein within the TGN [81,82]. Replication of SARS-CoV-2 is inhibited by synthetic furin inhibitors [83]. After entering the host cell, the virus suppresses ACE2 expression, that in turn increases the expression of angiotensin II (Ang II). Ang II binds to its receptor, the Ang II receptor type 1, followed by modulation of gene expression of many inflammatory cytokines through NF-κB signaling. This molecular interaction augments activation of macrophages resulting in the elicitation of inflammatory cytokines and leading to acute macrophage activation syndrome or respiratory distress syndrome. Metalloproteases, such as the ADAM metallopeptidase domain 17, cause cleavage of these pro-inflammatory cytokines as well as ACE2 receptors, leading to their release as soluble molecules. 

This process causes loss of the protective role of the surface ACE2 protein and potentially aggravates SARS-CoV-2 pathogenesis [84]. Macrophages and monocytes in the mononuclear phagocyte pathway infected with SARS-CoV-2 express several pro-inflammatory chemokines and cytokines, a process essential for the elicitation of systemic and local inflammatory reactions called cytokine storms [85]. 

## 5. Coronaviruses and Phytochemicals

The health crisis created by Coronaviruses, in particular, the pandemic SARS-CoV-2, warrants the identification of new safe antivirals to mitigate disease. However, such discernment is challenging because of the uncertainty in the behaviour of SARS-CoV-2 which poses great difficulty in determining the efficacy of compounds for treatment and prophylaxis [86]. The emerging SARS-CoV-2 virus bears resemblance in molecular characteristics with the other two major Coronaviruses, the SARS-CoV and MERS-CoV, therefore plant compounds effectual against the latter two viruses could be useful drug candidates for prevention and therapy of Covid19 disease. There are a multitude of various plant products investigated against Coronaviruses, most of which have high selectivity index which makes them highly effectual and safe [87]. Phytomedicines have gained popularity in recent times on account of their therapeutic properties and fewer side effects in comparison to allopathic medicines. Phytochemicals are used as pharmaceuticals or as functional foods. Herbal plants, such as *Eucalyptus caesia*, *Rosmarinus officinalis*, *Artemisia kermanensis*, *Satureja hortensis*, *Mentha* spp., *Zataria multiflora* and *Thymus* spp., are typically rich sources of compounds such as phenolics [88,89,90]. 

## 6. Mechanistic Facets Regarding the Use of Phytochemicals against Coronaviruses

Coronaviruses attack the immune system of their hosts via their proteases: the papain like protease (PLpro) and 3CLpro. The SARS-CoV PLpro antagonizes innate immune reaction. The immune system of the host first identifies the viral RNA following which it recruits the adaptor antiviral signaling proteins in the mitochondria to transduce molecular signals to the kinase complex downstream to activate the transcription factors, IRF-3 and NF-κB. In the next step, these transcription factors stimulate the expression of IFN-*α* and IFN-*β* type I interferons, following which STAT transcription factors are activated leading to the expression of IFN-stimulated genes (ISGs). This creates an antiviral state within the surrounding cells. The PLpro of SARS-CoV functions either by deubiquitinating proteins or by deISGylating, or both, within these molecular pathways resulting in the antagonism of the antiviral response in the host. This paves way for the successful evasion of the host immune response by the virus [91]. This implies that drug candidates capable of inhibiting the entry of the virus or replication or stimulating host immune response to synthesize type I IFN would be beneficial in treating SARS-CoV infection. By means of several investigations following the first SARS-CoV outbreak, it was found that herbal extracts and plant-based compounds function against viruses through various mechanisms. These mechanisms could be immunostimulatory, analgesic, inflammatory or by virtue of direct antiviral activity and modulation of symptoms. Direct antiviral activity includes blocking of early replication or viral entry [92] as well as inhibition of late replication [93]. 

## 7. Phytochemical Inhibitors of MERS-CoV

There are only a few studies investigating the use of phytochemicals as therapeutic compounds against MERS-CoV. A phytochemical, silvestrol from *Aglaia* sp., was identified to be a robust inhibitor of the replication of MERS-CoV with an EC50 value of 1.3 nM [94]. Silvestrol specifically inhibits RNA helicase eIF4A and therefore represses viral replication, precluding the formation of replication and transcription complexes, leading to the arrest of MERS-CoV protein expression. 

One of the most potent inhibitors of MERS-CoV is the lectin, griffithsin, found in the red alga *Griffithsia*. This compound contains three carbohydrate-binding domains that enables its binding to glycan moieties on the MERS-CoV protein spikes, thereby precluding the attachment of the virus to host cells. In vitro trials demonstrated the high potency of griffithsin against MERS-CoV with an EC50 value of ~0.125 μM [95]. Additionally, it exhibited diminished systemic toxicity and therefore appears highly promising as a primary drug candidate against MERS-CoV as well as other Coronaviruses. 

Polyphenols comprise a major class of phytochemicals having antiviral potential due to their abilities to block virus entry precluding viral infection at the early stages. A stilbenoid, resveratrol is an important natural product expressed in plants such as *Vaccinium macrocarpon*, *Polygonum cuspidatum* and *Vitis vinifera*. Resveratrol strongly inhibited infection by MERS-CoV and repressed replication of MERS-CoV in vitro. This compound could hence be considered as a robust anti-MERS agent and shows great promise towards application as an antiviral against SARS-CoV-2 [96]. 

## 8. Inhibition of SARS-CoV Using Phytochemicals

Inhibition of viral entry is a propitious area to be explored for the identification of appropriate drug candidates. Coronaviruses enter the cell through recognition and binding of the host cell receptor namely, angiotensin converting enzyme 2 (ACE2) wherein the virus S protein attaches to the ACE2 receptor. Several phytochemicals have demonstrated capability to strongly inhibit the interaction of the SARS-CoV S protein with the ACE2 receptor. For instance, anthraquinone compounds rhein and emodin as well as a flavonoid, chrysin, obtained from *Polygonum* and *Rheum* were studied for their anti-SARS-CoV activity. Amongst these compounds, emodin was found to be the strongest in inhibiting the S protein-ACE2 interaction showing an IC50 value of 200 μM [97].

Flavonoids as well as polyphenolic compounds such as tetra-*O*-galloyl-*β*-D-glucose, luteolin and quercetin were shown to strongly inhibit the cellular entry of the SARS-CoV [98] with EC50 values of 4.5 μM, 83.4 μM and 10.6 μM, respectively. The human immunodeficiency virus (HIV)-luc/SARS pseudo-typed virus was investigated for viral entry inhibition using the above three compounds wherein all of these compounds proved to inhibit viral entry, amongst which quercetin was found to have lowered cytotoxicity. Hence, quercetin could be considered as a prospective drug against SARS-CoV-2 and could be promising as an FDA-approved drug. 

Licorice roots contain several bioactive compounds inclusive of antivirals, anti-tumorals and anti-inflammatory compounds. Cinatl et al. [99] reported that glycyrrhizin, a saponin expressed in Licorice roots, can block the replication of SARS-CoV with an EC50 value of 300 to 600 mg/L, respectively, during and following virus adsorption. The CC50 value has been reported to be over 20,000 mg/L. This implies that, at this dosage, there occurs only 20–30% decrease in cell viability, making the compound less cytotoxic. Glycyrrhizin at 4000 mg/L completely inhibited virus replication in infected cells. This investigation also found that glycyrrhizin is safer when compared to other antivirals such as ribavirin which causes hemolysis and drastic depletion of hemoglobin in SARS-CoV patients. 

Glycyrrhizic acid and its aglycone, glycyrrhetinic acid are effective against a wide range of viruses such as flaviviruses, herpes viruses, HIV, Hepatitis C virus and importantly, SARS-CoV as discerned by in vitro studies. Inhibition of these viruses has been reported in Vero cells and in infected individuals [100,101,102]. Further investigations revealed that replacement of sugar moieties in glycyrrhizic acid led to loss of anti-SARS-CoV activity and therefore its effectuality as an antiviral depends on the presence of its sugar component. During the outbreak of SARS in 2002, glycyrrhizin characteristics and its mechanism of action were not deciphered. Subsequently, studies by Hoever et al., 2005 [103] demonstrated that, when 2-acetamido-β-D-gluco-pyranosylamine was added to the glycoside chain of glycyrrhizin, this resulted in a 10-fold higher activity of glycyrrhizin against SARS-CoV and blocks its entry into cells.

Lycorine, an alkaloid, was fractionated and purified from the *L. radiata* plant extract and this compound showed an EC50 value of 15.7 ± 1.2 nM as well as a CC50 value of 14,980.0 ± 912.0 nM as demonstrated in cytotoxicity assays. Lycorine inhibited virus replication at concentrations less than glycyrrhizin and could be considered as a potential candidate for generating new drugs against SARS-CoV [104]. SARS-CoV replication can be inhibited by disrupting the major viral protease (Mpro or 3CLpro). Among these, the 3CLpro controls the activities of the virus replication complex and is an important drug target [105]. SARS-CoV replication has also been disrupted by the alkaloid, reserpine obtained from the dried roots of Indian snakeroot, *Rauvolfia serpentine* and by the saponin, aescin obtained from the European chestnut, *Aesculus hippocastanum,* which respectively showed EC50 values between 3.4 μM and 6.0 μM as well as CC50 values of 25 μM and 15 μM. Both compounds interfered with virus entry and also successfully inhibited the 3CLpro enzyme activity within the cell [106]. 

Additionally, plant lectins have been demonstrated to strongly inhibit SARS-CoV entry into cells. Lectins obtained from various plants showed anti-SARS-CoV activity having an EC50 value ranging between 0.45 to >100 μg/mL as well as CC50 value between 50 to 100 μg/mL. Amongst these lectins, the highest antiviral activity was demonstrated for mannose-binding lectins obtained from *Allium porrum,* having a selectivity index of >222 [107]. This study showed that the lectins interfered with both viral entry and viral replication. 

Flavonoids have also been reported to exhibit antiviral activity against Coronaviruses. The flavonoids, including quercetin, luteolin and apigenin purified from ethanolic extracts of leaves from the Asian traditional medicinal plant, *Torreya nucifera,* showed potent inhibition of the SARS-CoV 3CLpro activity [108] wherein IC50 values of 23.8 μM, 20.2 μM and 280.8 μM were reported, respectively. Amongst these, a carbon moiety present in apigenin displayed stronger activity when compared to the other flavonoids. 

Likewise, terpenoids were also shown to be effectual in inhibiting viral replication. Pinusolidic acid, α-cadinol and ferruginol were purified from *Chamaecyparis obtuse* ethyl acetate extracts, betulonic acid and cedrane-3β,12-diol purified from *Juniperus formosana* plants and crypto-japonol purified from *Cryptomeria japonica.* These investigations showed that a majority of the terpenoids inhibited SARS-CoV replication at EC50 values between 3.8 and 7.5 μM and CC50 values more than 250 μM. Through ELISA assays, it was found that the terpenoids blocked the SARS-CoV S protein to inhibit the replication process. This study [109] also demonstrated that the terpenoids displayed potent anti-3CLpro activity and therefore, in their purified from, these terpenoids are favorable drug candidates to counter SARS-CoV infection. 

Besides inhibition of virus replication and 3CLpro activities, plant-derived compounds are effectual in inhibiting critical enzymes such as SARS helicase and 64 natural compounds in their purified forms were tested for their activity against SARS helicases. Of these, flavonoids scutellarein and myricetin, purified from *Aglaia perviridis,* demonstrated notable activities against SARS-CoV ([110]; Figure 1). Colorimetry-based ATP hydrolysis assay and double-strand DNA unwinding assay based on fluorescence resonance energy transfer (FRET) showed that these flavonoids strongly inhibited SARS-CoV helicase in vitro by impacting the ATPase activity. Scutellarein and myricetin displayed lower IC50 values at concentrations of 2.71 ± 0.19 μM and 0.86 ± 0.48 μM, respectively, and could therefore function well as biochemical inhibitors against SARS-CoV. 

Phytochemicals can also block other steps in SARS-CoV infection. Schwarz et al., 2011 [112] demonstrated that the anthraquinone, emodin, purified from plants belonging to the family *Polygonaceae* can block the coronavirus 3a ion channel and hence preclude the release of SARS-CoV virus from the infected cells. Besides these, phytochemicals have been effectual in anti-coronavirus immune modulation [113] and can also function as analgesics as well as in symptom regulation such as reduction of dyspnea and hypoxemia. Such studies suggest that these phytochemicals can also be used against SARS-CoV-2 infection. 

## 9. Plant-Based Phytochemical Therapeutics against SARS-CoV-2 Infection

*Senna L.* constitutes a major genus of legumes (*Fabaceae*) belonging to the subfamily *Caesalpinioideae*, comprising 300–350 species [114] that are both diverse and widespread. Several Senna species are typically used in herbal medicines and foods [115]. These plants contain, among others, important bioactive compounds such as terpenes, alkaloids and quinones [116,117,118,119,120]. Aqueous extracts of Senna leaves granted relief from SARS-CoV-2 symptoms in patients [88] and are being explored as a potential herbal therapy against SARS-CoV-2. 

A traditional Chinese medicine formula, Lianhuaqingwen (LH), containing a mixture of 13 herbs was demonstrated to inhibit SARS-CoV-2 replication, alter the morphology of cells infected with the virus and decrease the production of pro-inflammatory cytokines [121]. This LH formula, when used in in vitro studies, also showed equivalent antiviral potency. By transmission electron microscopy (TEM), it was demonstrated that there was a great reduction in the number of virus particles in SARS-CoV-2-infected patients when administered with LH at a concentration of 600 µg/mL. The exact mechanism of LH action is not yet clear although it has been reported to decrease cytokine release from virus-infected cells implicating the involvement of multiple levels of activities. Figure 2 shows some examples of phytochemicals capable of preventing SARS-CoV-2 infection. 

Health authorities in China have launched several herbal remedy programs to preclude SARS-CoV-2 infection and spread, major among which are the herbal formulas, Glycyrrhizae radix Et Rhizoma (Gancao) and Radix astragali (Huangqi) [123]. Additionally, tender leaf extracts of the popular Chinese vegetable, *Toona sinensis Roem* have been deemed to be used safely and have been shown to inhibit SARS-CoV in vitro [124]. It is probable that this extract can also be effective in inhibiting SARS-CoV-22. 

Previous studies showed that herbs capable of inhibiting SARS-CoV also successfully repressed SARS-CoV-2 infection. For instance, studies by Wen et al., 2011 [125] investigating 50 conventional Chinese medicinal herbs for anti-SARS-CoV activity using cytopathic assays based in Vero E6 cells revealed that six of these herbs could be used as probable SARS drug targets. Herbal extracts of the plants *Gentianae radix* (lóng dǎn), *Rhizoma cibotii* (gǒu jǐ), *Cassiae semen* (jué míng zǐ), *Dioscoreae rhizoma* (shān yào), and *Loranthi ramus* (sāng jì shēng) showed that all repressed SARS-CoV replication and two impacted SARS-CoV protease activity. Phlorotannins sourced from *Ecklonia cava*, an edible brown alga, showed that many of these compounds were successful in inhibiting SARS-CoV activity by impacting protease activity. One among these bioactive compounds, dieckol,1 demonstrated the highest antiviral activity. This occurred due to competitive binding within the catalytic site of the viral protease [126]. In this light, it is probable that dieckol could also function as an antiviral against SARS-CoV-2. 

## 10. Development of Vaccines and Therapeutic Interventions against Coronaviruses

Notwithstanding, by and large vaccination is the most viable strategy to preclude infectious diseases considering its specific immunity over long time periods and its ability to alleviate morbidity and mortality. Of all the proteins encoded by Coronaviruses, S protein acts as the principal antigenic determinant harboring neutralizing epitopes and therefore is considered as the prime target for design of vaccines. Particularly, specific antibodies against the S protein demonstrated highly immunodominant and enduring immune reactions against coronavirus infection [127,128]. Additionally, investigations in pre-clinical animal models have demonstrated that the S protein is capable of inducing robust cellular and humoral immune responses [129,130,131,132]. 

Several vaccine platforms have been adopted for developing MERS-CoV and SARS-CoV vaccines that are currently in the pre-clinical/clinical phases including inactivated, live-attenuated, viral vector, viral DNA, viral RNA and recombinant protein vaccines. Many vaccines against the MERS-CoV S protein, particularly the RBD domain,1 were formulated and tested in animal models inclusive of adenovirus vector, protein subunit and DNA vaccines. The GLS-5300 S protein vaccine based on viral DNA induced strong neutralizing antibodies in macaque models and protected against MERS-CoV infection, whereupon it was considered for clinical studies. This vaccine elicited immune responses as high as 85% when administered in two doses in human trials while showing no major adverse outcomes [133]. Likewise, recombinant MERS-CoV subunit vaccines based on the RBD showed enhanced immunogenicity in animal models by stimulating strong neutralizing antibodies and eliciting strong cellular immune reactions against MERS-CoV in addition to cross-neutralizing camel and human MERS-CoV strains with protracted immunity for 2as long as 6 months [134]. 

Recently, several Covid19 vaccines were generated using different platforms and have currently been authorized for emergency use inclusive of the inactivated SARS-CoV-2 vaccine, BBV152 (Bharat Biotech, Genome Valley, Hyderabad, India), CoronaVac (called PiCoVacc), a whole-inactivated SARS-CoV-2 vaccine produced by Beijing-based Sinovac Biotech company, China [135], BNT162b2 (BioNTech/Pfizer, Mainz, Germany) [136], mRNA-1273 (Massachusetts-based USA biotechnology company Moderna, Cambridge, MA, USA) [137], NVX-CoV2373, an S protein-based vaccine candidate developed by Novavax in Gaithersburg, MD, USA [138] and ChAdOx1 nCoV-19 (AZD1222), a non-replicating SARS-CoV-2 viral-vectored vaccine generated by AstraZeneca in Cambridge, UK [139]. 

## 11. Plant Molecular Pharming to Combat Coronavirus Infections

The production of important recombinant pharmaceuticals using plant cells/tissues cultured in vitro or using whole plants is called molecular farming [140,141]. Plant-based expression systems have been used for producing recombinant therapeutic proteins for nearly 3 decades [11,142,143]. Plant-based bioreactors/systems have several advantages such as low production and maintenance costs, safety, diminished risks of contamination due to human pathogens, tractability to implement desired post-translational modifications by engineering the plant glycosylation machinery, as well as amenability to being employed for manufacturing on a large scale [88,144]. Various methods including cell-based expression and stable and transient expression have been adopted for producing recombinant proteins in plants. 

Transient expression in plants is gaining increased interest in recent times due to its tractability and expeditiousness in generating large amounts of recombinant biopharmaceutical proteins such as monoclonal antibodies, vaccine candidates and diagnostic reagents to meet with demands imposed by pandemics and disease emergencies [145,146]. Several plant species, including *Nicotiana benthamiana* [131,147,148,149,150], *Nicotiana tabacum* [151,152], *Arabidopsis thaliana* [153,154] and *Lactuca sativa* [155,156] have been used as transient expression systems. Among these, *N. benthamiana* has been found to be the most favored transient expression platform. Generally, plants 4–6 weeks old are employed for transient expression using viral expression vectors derived from plant viruses such as alfalfa mosaic virus, cowpea mosaic virus, potato virus X, plum pox virus and tobacco mosaic virus, or using *Agrobacterium tumefaciens* carrying the recombinant expression cassette. 

Transient expression is highly favorable in comparison to other plant-based expression systems due to its ease of use, low-cost, rapidity and increased yield of recombinant protein products [157]. The respective genes of interest are infiltrated into the plants which then express the recombinant proteins within a span of 3–4 days [158,159,160]. This enables efficacious expression of several self-assembling viral antigens at high levels. 

Vaccine antigens expressed in plants have been demonstrated to elicit potent immune reactions in humans. Several vaccine antigens and virus-like particles have been engineered using plant-based bioreactor systems. Plants, upon transformation with foreign genes, generate a plenitude of vaccine antigens, antibodies and drugs that can be used to combat various human pathogens, thus making them safe, low-cost and trouble-free reservoirs for storage of vaccine proteins and drugs [161]. 

The S1 protein of SARS-CoV has been stably expressed in low nicotine tobacco plants and tomato plants which reportedly induced antibody response in murine models [162]. Transient expression of the recombinant N protein of SARS-CoV in N. benthamiana demonstrated high immunogenicity, in particular humoral immune response [163]. Knowledge gained from previous investigations can aid the design and development of efficient plant-derived vaccines against the currently emerging pathogen SARS-CoV-2 [164]. The virus sequence of SARS-CoV-2 was published in early 2020 and ever since several advancements have been made in generating plant-based recombinant vaccines against SARS-CoV-2. Table 2 lists some of the plant-based vaccine candidates to combat coronavirus infections.

To-date there are 227 vaccine candidates against SARS-CoV-2 in clinical trials, including 47 vaccines already approved and 89 vaccines in phase 3 clinical trials (trackvaccines.org). The pioneering production of a successful plant-based vaccine against SARS-CoV-2 was reported by Medicago, a Canadian biopharmaceutical company [173]. In this strategy, the gene sequence of the modified S protein of SARS-CoV-2 containing stabilizing point mutations was introduced into *Agrobacterium* followed by transformation of the plant, *N. benthamiana* with the engineered *Agrobacterium* [88,157]. The S protein was expressed along with a plant signal peptide as well as the transmembrane domain and cytoplasmic tail of influenza HA in the place of analogous sequences in SARS-CoV-2 and this was expressed transiently in *N. benthamiana,* wherein the S protein assembled into VLPs. The resulting transgenic/recombinant plants express VLPs which are composed of the SARS-CoV-2 spike protein and the plant lipid membrane, wherein these VLPs exhibit size and shape analogous to the actual SARS-CoV-2, but bereft of any genetic material, thus rendering them non-infectious [174]. These Covid19 VLPs were formulated with AS03: GSK or CpG 1018: Dynavax as adjuvants. Of these, AS03-adjuvanted vaccine candidates stimulated stronger immune responses than that of the CpG1018-adjuvanted vaccine formulations and there were no adverse reactions. Interferon and interleukin-4 cellular responses specific to the S protein were observed. After successful results obtained in their Phase-1 studies, Medicago launched Phase-2/3 (ClinicalTrials.gov Identifier: NCT04636697) as well as Phase-3 clinical trials (NCT05040789) [173,175]. Recently, GlaxoSmithKline and Medicago Inc., reported a 71% rate of efficacy for their plant-derived vaccine against all SARS-CoV-2 variants [Medicago Inc., 2022]. Medicago’s VLP-based SARS-CoV-2 vaccine is estimated to reach the expression capacity of as high as 10 million doses on a monthly basis. The success of Medicago’s SARS-CoV-2 vaccine [176] comes in the wake of their propitious development of the hemagglutinin-based influenza virus VLP vaccine [177] which has shown tremendous success in human clinical trials in terms of its efficacy and safety in administration. The generation of such VLP-based vaccines costs only a small fraction when compared to their conventional equivalents [178]. 

The American biotechnology company Kentucky BioProcessing [173] announced the production of a plant-based SARS-CoV-2 subunit vaccine. KBP used *N. benthamiana* plants for transiently expressing their KBP-201 vaccine. The KBP-201 vaccine is presently in phase1/2 clinical trials [179] wherein it is being administered along with CpG oligonucleotides that function as adjuvant (ClinicalTrials.gov Identifier: NCT04473690). As high as 1–3 million doses of this vaccine can be generated per week. Similar plant-derived approaches have been used to produce 10 million doses of the flu vaccine and the Ebola vaccine within one month [180]. 

Cape Bio Pharms (CBP), a South African company, has generated SARS-CoV-2 diagnostic kits [181] in which various regions of the SARS-CoV2 S1 glycoprotein were expressed as fusion proteins. Additionally, they are generating antibodies against these proteins in collaboration with antibody manufacturers [181]. In Canada, Suncor and the University of Western Ontario have developed algal systems to express SARS-CoV-2 spike proteins for designing SARS-CoV-2 diagnostic test kits [182]. Algae serve as superior bio-factories owing to their facile growth requirements and ease of modification/engineering to express viral proteins.

Another company, iBio from Bryan, Texas, is generating IBIO-200, a VLP-based vaccine, IBIO-201 which is a SARS-CoV-2 spike-based subunit vaccine in combination with LicKM™ booster molecule, and IBIO-202 which is a subunit vaccine candidate targeting the SARS-CoV-2 nucleocapsid protein. The LickKM and FastPharming technologies were coupled in their manufacturing process using subunit vaccine and VLP platforms to generate the Covid19 vaccine [174]. The emergence and re-emergence of mutated strains of Coronaviruses has raised apprehensions and diverted this biotechnology company to develop the second-generation IBIO-202 vaccine that affords broader protection using the highly conserved sequences of the nucleocapsid protein of SARS-CoV-2 harboring immunogenic epitopes, considering that the emerging new variants would be less capable of escaping vaccine protection [183,184,185,186]. This vaccine has been proved to trigger immune responses against SARS-CoV-2 in preclinical investigations [187]. The preclinical trial of IBIO-202 was completed in July 2021 while the IBIO-201 preclinical trials were also completed with no adverse outcomes at both low and high doses [168,169]. 

Another subunit vaccine against SARS-CoV-2 was developed in *N. benthamiana* by Baiya Phytopharm from Thailand using the protein expression platform, BaiyaPharming™. Six candidate vaccines were studied for their efficiency and, based on the outcomes, one candidate Baiya SARS-CoV-2 Vax 1 was chosen that demonstrated augmented immunity in monkeys and mice. Following this, further efficacy and safety studies were conducted with the intent to commence Phase I clinical trials from September 2021 (ClinicalTrials.gov Identifier: NCT04953078) [188]. 

The Lomonosoff research team at the John Innes Center, Norwich, UK, recently announced the production of a SARS-CoV-2 vaccine candidate based on the viral S, M and E proteins in plants. They demonstrated the purification of crown-shaped particles from leaves of plants infiltrated with the virus S protein [189]. The laboratory of Nicole Steinmetz at the University of California, San Diego, has generated Cowpea mosaic virus VLPs displaying T- and B-cell epitopes of the SARS-CoV-2 S protein on their icosahedral surfaces [190]. This recombinant virus can be used for parenteral administration through an implanted microneedle technique to elicit potent immune reaction against SARS-CoV-2 [191]. This group has also developed augmented Covid19 diagnostic kits using this technology, wherein the recombinant CPMV VLPs have been used as positive control probes. These VLPs show stability at room temperature for long time periods and can be generated at low-cost which makes them ideal systems for administration in resource-poor settings [192]. 

Yet another collaborative effort between two research teams in the University of Toronto, Canada, has presented a novel method to combat SARS-CoV-2 that uses a synthetic peptide capable of interacting with the viral deubquitinase (DUB) and is harbored in a plant virus [88]. This synthetic peptide, called ubiquitin variant (UbV), contains approximately 80 amino acids and blocks the deubiquitinase as well as the proteolytic activities of the virus, thus precluding viral infection [193]. Ubiquitin variants of both MERS-CoV and SARS-CoV-2 viruses have been engineered wherein their respective UbV peptides have been synthesized as fusions with the N-terminus of the Papaya mosaic potex-virus (PapMV) capsid protein which upon assembly into VLPs within plant cells, can be used to preclude the establishment of infections by these viruses. PapMV has been earlier demonstrated to enter into human cells through the cytoskeletal protein, vimentin [88]. Therefore, PapMV VLPs loaded with SARS-CoV-2 UbV can gain entry into cells and inhibit virus infection. Nanoparticles of potex-viruses have been reported to successfully infiltrate lung epithelial cells upon administration as an aerosol spray. Hence, the PapMV VLPs carrying the SARS-CoV-2 UbV variant can be impregnated into an inhaler that can be used for the treatment of the lungs of infected patients. 

This UbV variant is also being expressed in a plant geminiviral vector to enable its purification as a SARS-CoV-2 antiviral [88]. Gemini-viruses such as the Bean yellow dwarf virus have been genetically engineered to generate large quantities of pharmaceutical proteins in plants in relatively brief time periods [194]. Additionally, a novel synthetic SARS-CoV-2 antibody engineered through a phage display library is presently being considered for overexpression using this geminiviral vector system [145]. 

Probable antiviral targets can be engineered as antiviral drugs and vaccines against different viral proteins such as the spike, envelope, membrane, nucleocapsid, virus-encoded RNA polymerase and the 3-chymotrypsin-like protease (3CLpro) that cleaves the SARS-CoV-2 polyprotein at 11 sites, in order to enable the generation of viral non-structural proteins essential for viral replication [88]. VLPs of SARS-CoV-2 seem to be propitious vaccine candidates by virtue of their potential to stimulate immune responses in a manner analogous with that of the natural virus. 

## 12. Plant-Derived Monoclonal Antibodies, Therapeutic Proteins and Diagnostic Reagents against Coronaviruses

Monoclonal antibodies (mAbs) have been employed as therapeutic molecules to treat many human diseases and have emerged as a major class of biopharmaceuticals in recent times [195]. The application of plant-produced mAbs circumvents the shortcomings involved in passive immunizations, in particular intravenous immunoglobulin or serum immunotherapy, in terms of functionality, specificity, purity and safety due to the lowered risk of contamination with human pathogens [7,196]. 

Therapeutic mAbs with potential to combat coronavirus infections have recently been identified. These mAbs target the S protein or the RBD demonstrating high anti-viral potency and significantly diminishing viral load by affecting the binding of the RBD with its receptor on the cell surface [197,198,199]. Additionally, these S-specific mAbs preclude the fusion of the viral and cellular membranes and thus block the entry of the virus and the subsequent infection [200]. Therefore, the development of neutralizing antibodies specific to CoV S or RBD can act as an efficacious strategy to enable passive immunity. A plethora of mAbs possessing therapeutic potential have been reported for MERS-CoV [201,202], SARS-CoV [198,203,204] and for SARS-CoV-2 [205,206,207,208,209,210,211,212,213].

Aside from vaccine antigens, plants have been shown to be efficient systems for rapid production of monoclonal antibodies to thwart viral infection. Plant-derived monoclonal antibodies and protein-based subunit vaccines have been developed against several emerging diseases such as Covid19 [214]. Plants have the capability to produce fully assembled and functional mAbs for veterinary and human applications. Passive immunization mediated by mAbs can provide immediate protection against viral infections. 

Recently, a proof-of-concept study was conducted to demonstrate the efficacy of plant expression systems to generate anti-SARS-CoV/CoV-2 mAbs for passive immunotherapy [205]. Using a geminiviral vector, the human anti-SARS-CoV-2 H4 and B38 mAbs were produced in *N. benthamiana*. The heavy and light chain genes of the mAbs were co-expressed transiently to obtain yield of 4 to 35 µg of mAb/g FW in the leaves. Both the plant-generated B38 and H4 mAbs showed antigen binding specificity and neutralizing anti-SARS-CoV-2 activity in vitro studies [205].

Diego-Martin et al., 2020 [167], expressed six antibodies against SARS-CoV-2 in *N. benthamiana* and reported expression levels between 73 and 192 µg/g FW. Additionally, the plant-generated SARS-CoV2 antigens have been successfully used as diagnostic reagents to manufacture rapid test kits to detect SARS-CoV-2 infection. Makatsa et al., 2021 [215], expressed recombinant S1 and RBD polypeptides of SARS-CoV-2 in plants which successfully detected antibodies specific to SARS-CoV-2 in sera of patients who were diagnosed as positive by PCR analysis. 

Baiya Phytopharm also developed an IgM/IgG test kit against Covid19 containing a lateral-flow immunoassay strip using the recombinant SARS-CoV-2 RBD produced in plants. In total, 51 serum samples confirmed for Covid19 infection were examined using this strip wherein the specificity and sensitivity of the kit was shown to be 98% and 94.1%, respectively [216]. Table 3 enlists the plant-based mAbs and coronavirus diagnostic reagents expressed in plants. 

In recent studies, a fusion protein containing the SARS-CoV-2 RBD sequence and the immunoglobulin Fc region has been transiently expressed in *N. benthamiana* plants. This plant-based RBD-Fc fusion protein induced broad cellular and humoral immune responses when administered intramuscularly in cynomolgus macaques and mice. Additionally, the effectiveness of adjuvants in augmenting the immune responses of plant-based anti-SARS-CoV-2 subunit vaccines was also demonstrated [129,170]. 

## 13. Additional Features of Plant-Based Vaccines against Coronaviruses

Generation of plant-based vaccines by transient expression enables them to be rapidly scaled up to meet immediate and unanticipated demands, which makes them an appropriate platform for vaccine production in the events of pandemics like that which we are presently experiencing. The success of this strategy is proved by many molecular farming companies specializing in the generation of plant-based proteins such as Ventria Bioscience/Invitria (Fort Collins, CO, USA), Fraunhofer (Newark, DE, USA), Medicago Inc. (Quebec City, QC, Canada), Agrenvec (Madrid, Spain), ORF Genetics (Kópavogur, Iceland), Diamante (Verona, Italy) and Protalix BioTherapeutics (Karmiel, Israel) [219]. 

Subunit vaccines designed using individual proteins or VLPs wherein multiple copies of the viral antigens are arrayed on the VLP surface have been produced [220]. The prototype vaccine against SARS-CoV-1 was successfully generated in 3 weeks after obtaining the protein sequences, and up to 200 mg of protein was expressed per kg of leaves FW [162]. The VLP technology has multiple advantages over subunit vaccines as the ordered arrangement of the vaccine antigen can elicit more potent humoral and cellular immune responses. 

Therefore, VLPs constituted by the SARS-CoV-2 S proteins that are generated by transient expression could be considered as the primary choice in the application of plant biotechnology to confront the COVID pandemic. As an alternative to the generation of VLPs composed of entire proteins or their truncated versions, key virus epitopes could be displayed on the VLP surface and thus act as vaccine immunogens. Such epitopes could be attached covalently or non-covalently to carrier VLPs, in particular those assembled using the Hepatitis B virus core Antigen (HBcAg) [221,222]. Such mono-epitopic VLPs would be composed solely of HBc-epitope monomers. The mosaic structure of the VLPs can further enhance the stability of the VLPs presenting heterologous surface epitopes. Such epi-topic VLPs decrease steric hindrance due to particularly large epitopes or those having specific structures. Other than HBcAg, S-HBsAg (Small Hepatitis B surface antigen) and the HPV (Human Papilloma Virus L1 protein) have been used as VLP carriers [223,224,225]. 

Recently, VLPs assembled from plant viruses such as CPMV, TMV, PVX or PapMV have been used as epitope carriers or for other nanotechnology and biopharming purposes [226,227]. Such plant virus VLPs can be safely expressed in substantial amounts due to the capability for large-scale proliferation of the plant viruses in appropriate hosts [228]. Plant virus VLPs could additionally confer adjuvanting properties and therefore could be the ideal option for combating emerging CoV mutants. Such VLPs can be purified and administered parenterally or can be delivered via the mucosal route such as sublingual, inhaled or intranasal vaccines. Alternatively, edible or oral vaccines can be generated using stable trans-plastomic or transgenic expression in plants and this strategy only requires partial processing of the plant tissue, typically by lyophilization [229]. Notwithstanding, these plant-derived edible vaccines necessitate the development of appropriate administration regimens. 

Hager et al., 2022 [230] report one of the first plant-based vaccines, CoVLP+AS03, approved for use in humans. The purification and downstream processing procedures for this vaccine are similar to all other recombinant vaccine schemes. However, the upstream protocols for such plant-derived vaccines are dependent only on sunlight, tightly regulated use of water and substrates for growth to sustain the living plant ‘bioreactor’. Plant-produced vaccines that target new virus variants can be manufactured within only a few months on a large scale [231]. The potential impact of this plant-based approach to counter the current pandemic will be largely influenced by the evolvement of the SARS-CoV-2-induced COVID-19 pandemic itself. Nevertheless, the availability and future advancement of this platform would have significant implications on the readiness for this and other pandemics. 

The CoVLP reported in the study by Hager et al., 2022 [230] shows stability at refrigerator temperatures, enabling ease of use in resource-poor countries [231]. This CoVLP+AS03 vaccine could also be administered as a booster following primary immunization with other vaccination schemes, for which studies are already under way. This CoVLP vaccine has been found to be stable for a minimum of 6 months at 2-8C. The adjuvant system (AS03, GlaxoSmithKline) primes a transient innate immune response [232] and enhances the quality, magnitude and endurance of the adaptive responses [233]. Preliminary studies show that the CoVLP+AS03 vaccine (Covifenz, Medicago) elicited potent and persistent levels of neutralizing antibodies, as well as a balanced T-cell immune response (interleukin-4 and interferon-gamma) [173,234]. The initial outcomes of the pivotal phase 3 trial for this vaccine have been reported, evaluating its safety and efficacy [230]. 

Transient expression of vaccine candidates involves agroinfiltration wherein foreign genes are introduced into plant leaves using infiltration with disarmed *Agrobacterium tumefaciens* harboring binary vectors [235]. This plant-based technology has enabled the large-scale and rapid expression of vaccines and therapeutics [236]. On the other hand, stable transformation using the transgenic technology requires longer time than agroinfiltration, but allows the long-term, large-scale production of the recombinant vaccine candidate. Several plant species such as tomato, cereals, carrot, tobacco, potato, lettuce, oil seeds and alfalfa have been utilized for stable transformation by manipulation of chloroplast or nuclear genomes, respectively, using biolistic guns or Agrobacterium-mediated transformation [157,237]. Chloroplasts are at times a more preferred target for such stable transformation rather than the nuclei, owing to their high copy number within the plant cell, and therefore lead to higher level expression of the recombinant vaccine. 

The technology of choice with respect to a plant-based vaccine is dependent on the route of administration, the capability to produce elevated levels of the recombinant vaccine and requirement of minimal downstream processing [238]. Edible plants such as rice, maize, lettuce, carrot and tomato are useful for oral administration. On the other hand, cereal crops are preferred for long-term storage [238]. Tobacco plants are suitable as model plants and enable increased levels of expression. Bioreactors using plant cell cultures for plant molecular farming helps circumvent the disadvantages of transgenic crops. Growth of plant cells in vitro under regulated conditions enables the precise inspection of cell growth and protein expression and the formulation of good manufacturing practices. The strategy of expressing important pharmaceutical proteins in plant cell bioreactors is becoming more common due to the strong containment measures on the cultivation of transgenic plants at the field level [239]. 

Endeavors to express plant-based SARS-CoV-2 VLPs appropriate for vaccine development in a brief time period are unprecedented [61,169,176]. Pogrebnyak et al., 2005 [162], successfully generated the S1 protein of SARS-CoV containing the receptor-binding domain and the N-terminal domain in tomato and tobacco plants, which showed strong immune responses when studied in murine models. Yet another investigation showed S1 protein expression in lettuce as well as trans-plastomic tobacco, but nevertheless, were not tested for elicitation of immune responses [165]. *N. benthamiana* was used to express the SARS-CoV N protein by Zheng et al. [163], while the N and M proteins were expressed by Demurtas et al. [166], wherein both vaccine candidates proved to be immunogenic in animal models. The receptor-binding domain (RBD) and N protein of SARS-CoV-2 were expressed in *N. benthamiana,* which elicited high titer of antibodies against the virus [240]. The RBD of SARS-CoV-2 produced in plants showed elicitation of humoral immune reactions in mouse models [241]. 

The rapid emergence of virus variants and the prospect of viral immune evasion reinforce the compelling need to develop vaccines that provide broad protection. Dube et al., 2022 [242] demonstrated a recombinant plant-based VLP vaccine for the ancestral SARS-CoV-2 (CoVLP) which has recently been authorized by health authorities in Canada. The modified CoVLP.B1351 formulated with the AS03 adjuvant has been used to target the B.1.351 variant in mouse models involving heterologous and homologous prime-boost regimens. Both approaches were shown to induce potent, broadly cross-reactive neutralizing antibodies against many variants of concern including the B.1.1.7/Alpha, B.1.351/Beta, P.1/Gamma, B.1.1.529/Omicron and B.1.617.2/Delta strains. The neutralizing antibody reactions were robust for both vaccination regimens although higher against almost all variants after the heterologous prime-boost regimen. 

## 14. Recent Developments Regarding Intervention Strategies against Coronaviruses

Edible nanoparticles (ENPs) are dietary plant-based exosome-like vesicles. Kalarikkal and Sundaram, 2021 [243] identified 22 ENP-derived miRNAs that could conceivably target different regions of the SARS-CoV-2 genome. 11 miRNAs displayed absolute specificity towards the SARS-CoV-2 target, but not SARS-CoV. ENPs sourced from ginger, grapefruit, hamimelon, pear, soybean and tomato contained multiple miRNAs targeting various regions of SARS-CoV-2. Intriguingly, the osa/cme miR-530b-5p showed targeting specificity towards the site of ribosomal slippage between the ORF1a and ORF1b in the SARS-CoV-2 genome. Further, the relative expression levels of six of the miRNAs (miR-169, miR156a, miR-166 m, miR-5077, miR-5059 and miR-6300) in grapefruit and ginger ENPs were validated by RT-PCR which displayed differential accumulation of specific miRNAs in these plants. Since ENPs upon administration led to their accretion within lung tissues in in vivo studies, the ENP-derived miRNAs specifically targeting the SARS-CoV-2 genome have great potential to be formulated as an alternate therapeutic agent against Covid19. 

Kanjanasirirat et al., 2020 [244] reported that extracts from *Boesenbergia rotunda* and its phytochemical, panduratin A displayed strong activity against SARS-CoV-2. Treatment with panduratin A and the extract following viral infection drastically inhibited SARS-CoV-2 infectivity as demonstrated in Vero E6 cells wherein IC50s of 0.81 μΜ (CC_50_ = 14.71 µM) and 3.62 μg/mL (CC_50_ = 28.06 µg/mL) were observed, respectively. Further, administration with panduratin A at the SARS-CoV-2 pre-entry phase blocked the infection with an IC_50_ of 5.30 µM (CC_50_ = 43.47 µM). This study is the first to report the inhibitory effect of panduratin A on both the pre-entry and post-infection phases of SARS-CoV-2 infection. Treatment with panduratin A also suppressed SARS-CoV-2 infectivity within human airway epithelial cells. This result substantiated the potential of panduratin A as an antiviral agent against SARS-CoV-2. As *B. rotunda* is used as a culinary herb in Southeast Asia and China, extracts from this plant as well as panduratin A in its purified from could be considered as potential candidates for effectual, low-cost treatment options during the Covid19 crisis. 

Kim et al., 2022 [245] reported that phytochemicals Jejuguajavone A, β-caryophyllene, pheophytin a and pheophytin b, sourced from the edible plants, *Psidium guajava* and *Chlorella* spp., exhibited broad-spectrum activity against several SARS-CoV-2 variants. The antiviral capabilities of these compounds were evaluated in vitro using Vero cell lines and in vivo using animal models such as hACE2 transgenic (TG) mice and golden Syrian hamsters. Intriguingly, synergistic anti-proinflammatory and antiviral activities were observed in animals receiving oral doses of a combination of these compounds. Kim et al., 2022 [245] thus showed that plant-based novel bioactive substances exhibit strong antiviral characteristics against a wide range of SARS-CoV-2 strains, as well as their variants, and phytochemicals obtained from edible plants could be potential candidates for producing rapidly available, inexpensive antiviral prophylactics and therapeutics. Mixtures of pheophytins with Jejuguajavone A or β-caryophyllene provided as a combinatorial treatment also displayed notable synergistic effects against the virus leading to the protection of SARS-CoV-2 infected animals from lethal challenge with viral variants inclusive of Omicron. The antiviral potential of combinations of these compounds could likely be due to the manifestation of multicomponent and multi-target drugs (MCMT), providing substantial evidence of its efficacy against SARS-CoV-2 inclusive of the recently emerging variants. SARS-CoV-2 delays innate immune reactions and hampers the immune system, leading to unbridled and aggravated inflammatory responses (Coperchini et al., 2020; Csermely et al., 2005; Kasuga et al., 2021) [246,247,248]. Myeloid-derived immune cells such as macrophages induce ‘cytokine storm’ driven by proinflammatory chemokines and cytokines during SARS-CoV-2 infection (Pedersen and Ho 2020; Zhang et al., 2021) [249,250]. The combination of the above stated compounds attenuates proinflammatory response induction and generation of pro-inflammatory chemokines and cytokines in a dose-dependent manner, while upregulating IFN-γ. This suggests that these phytochemicals can be used to mitigate SARS-CoV-2 infection by the suppression of the TLR3 pathway. Contrary to pheophytins, Jejuguajavone A and β-caryophyllene cause additional inhibition of chemotaxis that could disrupt the migration of proinflammatory cells to the infection site. Hence, the immunomodulatory and anti-inflammatory characteristics of these compounds in addition to their antiviral effects indicate that they could be considered as additional therapeutic interventions against Covid19. Recent outbreaks of the Omicron and Delta variants show that there is an urgent and compelling need for the development of low-cost, readily accessible Covid19 prophylactics and therapeutics, particularly in developing nations [245] (Kim et al., 2022). 

The medical field has transformed from totally “synthetic” to “semi-herbal” in recent times. Due to a dearth in effective therapeutic and disease management systems against SASR-CoV-2, alternative treatment measures are being investigated. Conventionally practised drug development schemes involve lengthy, time-taking regimens in addition to its inability to generate drugs as required. Furthermore, the SARS-CoV-2 is highly mutable with varying reproduction numbers (Rahman et al., 2020) [251] at a higher degree compared to the MERS-CoV and SARS-CoV (Liu et al., 2020) [252], because of which drug development against these viruses is challenging. Drugs targeting conserved regions of the viral genome or proteins encoded by these regions such as the spike protein or the major protease are likely therapeutic candidates. 

Amongst these measures, drug repurposing is the most favorable strategy. Typically, great time and effort are necessary to develop efficacious, safe drugs. Drug repurposing is deemed to be a promising strategy to overcome the urgent need for prophylactics and therapeutics to combat the Covid19 pandemic. Using the drug repurposing strategy, the search for Covid19 cure can be greatly expedited by dispensing with the need for foregoing phase I and phase II clinical trials dedicated towards the establishment of the dosage and safety of the respective drugs. This circumvents the time constraints of the regulatory procedures necessary to launch a new drug into the market and reduces the scope of comprehensive exploration for the most appropriate cure. Despite the fact that antiviral drugs and vaccines against SARS-CoV-2 are now available, effective prevention and containment of Covid19 disease through such interventions is circumscribed by inaccessibility and high costs in resource-poor settings especially in developing countries. Additionally, incomplete clearance of the virus due to ineffective therapeutics could result in the emergence of new viral variants capable of escaping the immune system of the host and the currently available SARS-CoV-2 vaccines. 

In spite of the unprecedently rapid development of SARS-CoV-2 vaccines, the emergence and re-emergence of novel viral variants poses a continued threat to human health the world over. Additionally, ineffectual control of virus spread, and unbalanced dissemination of therapeutics and vaccines have allowed the Covid19 pandemic to go on an infectious rampage across large populations. The emergence of new variants resulting from rapid evolution of SARS-CoV-2 has led to rapid spread globally, as observed with the recent Omicron variant outbreak in Africa (Sahoo and Samal, 2021) [253]. Besides, these emerging variants could cause reverse zoonotic infections in animals, likely leading to the establishment of reservoirs of these novel viral variants [254,255,256] (Halfmann et al., 2020; Schlottau et al., 2020; Tiwari et al., 2020). Therefore, to circumvent these impediments, there is a compelling, urgent need for broad spectrum and readily available antiviral agents. As Coronaviruses contain the largest genome in the kingdom of RNA viruses and code for diverse viral proteins associated with viral pathogenesis and host immune evasion, multicomponent and multi-target drugs can alleviate pathogenesis induced by Coronaviruses. 

Together with molecular dynamics and molecular docking studies, medicinal chemistry can be employed to discover potent candidate phytochemical compounds that can interact effectually with the active site groove of viral catalytic proteins, thus facilitating target inhibition (Rakib et al., 2021; Mahmud et al., 2021) [257,258]. Most of the investigations regarding the identification on natural compounds in the treatment of Covid19 disease include computer docking in silico studies to predict the anti-CoV effects of these substances as well as in vitro and in vivo screening of the efficacy of these phytochemicals against the CoVs (Mani et al., 2020; Zhang D.-h. et al., 2020) [102,259].

Computer-assisted drug design involves the creation of an artificial environment simulating the human body environment (Poojary, 2020) [260] which enables the examination of molecular interactions between bioactive compounds and a given target protein under simulated physiological conditions in a cost-efficient and time-saving fashion. By means of in silico analysis, commercially produced drugs are docked with a viral protein target and these screened drugs could be made accessible for patients within a much lower timeframe as the clinical profile of such drugs is already well established. Numerous compounds can be screened against a particular target protein in the virus to narrow down the number of compounds to be examined in vitro using bioinformatic techniques. The strategy of molecular docking helps in evaluating and visualizing the binding interactions between candidate ligands and the target proteins. Some drugs such as Lopinavir, Ribavirin, Ritonavir, Remdesivir, Darunavir, Favipiravir, hydroxychloroquine, chloroquine, type I and type II interferons, Tocilizumab, statins and arbidiol, have been repurposed as antiviral drugs for SARS-CoV-2 (Singh et al., 2020) [261]. However, despite this approach enabling speedy antiviral effects, several in vivo and in vitro studies are necessary to fully discern their mechanisms of action within the human body, particularly when issues of elevated comorbidities are encountered. The unfavorable side-effects of these synthetic drugs have compelled scientists and researchers towards generating plant-based medicines. Several compounds extracted from plants belonging to families such as *Asteraceae*, *Fabaceae*, *Lamiaceae*, *Geraniaceae*, *Malvaceae*, *Rutaceae* and *Rosaceae* have been demonstrated to show anti-SARS-CoV-2 antiviral activity (Drevinskas et al., 2018; Denaro et al., 2020; Siddiqui et al., 2020) [262,263,264]. 

Using molecular docking, several potential SARS-CoV-2 inhibitors such as Somniferine [9.62 kcal/mol] and Withanoside V [10.32 kcal/mol] from *Withania somnifera* (Ashwagandha), Tinocordiside [8.10 kcal/mol]) from *Tinospora cordifolia* (Giloy) as well as Ursolic acid [8.52 kcal/mol]), Isorientin 40-O-glucoside 200-O-p-hydroxybenzoagte [8.55 kcal/mol] and (Vicenin [8.97 kcal/mol] from *Ocimum sanctum* (Tulsi) were identified (Shree et al., 2022) [265]. Prediction using ADMET (Absorption, Distribution, Metabolism, Excretion and Toxicity) profiling revealed that these phytochemicals best docked in this study and possessed drug-like properties, while being non-toxic and therefore safe for administration. Subsequent MD simulation studies substantiated the stability of the docked complexes. Active phytochemicals obtained from these medicinal plants could potentially block SARS-CoV-2 MPro/3Clpro and therefore could be candidates for use in combating Covid19 disease. 

Chikhale et al., 2021 [266] performed molecular docking studies on Quercetin glucoside and Withanoside X obtained from *Withania somnifera* (Indian ginseng), which showed favorable binding interactions with the SARS-CoV-2 NSP15 endoribonuclease and the RBD of the prefusion S proteins. Withanoside X exhibited the highest binding free energy [Δ*G*_bind_ = −89.42 kcal/mol] and therefore could be considered the most promising candidate as an inhibitor. Taken together, these bioactive compounds from Indian ginseng proved to be successful inhibitors as per MD studies and could be an option as antiviral agents in combating SARS-CoV-2 infection. 

Singh et al., 2021 [267] used in silico approaches such as molecular dynamics simulations, molecular docking and end-state thermodynamics to screen and evaluate natural compounds from plants for their capability to repress the SARS-CoV-2 S protein RBD. From this, they could identify that Diacetylcurcumin and Dicaffeoylquinic acid had the potential to function as SARS-CoV-2 RBD inhibitors. Particularly, Dicaffeoylquinic acid was shown to bind strongly with essential amino acid residues (Lys417, Tyr489, Gln493, Tyr473, Glu484 and Phe456) of the S-RBD. Since these residues are associated in interactions between the S-RBD and the ACE2 receptor, Dicaffeoylquinic acid could potentially inhibit entry of the virus into host cells. 

Kulkarni et al., 2021 [268] reported the identification of 43 known compounds in the marine microalga *Ulva intestinalis* L. which were screened for docking against the SARS-CoV-2 S1 receptor binding domain (RBD), out of which compounds showing elevated binding affinity were selected for further pharmacokinetic studies. Results showed that several aliphatic compounds, phyto-steroids, polyenes and phenols obtained from the extract including retinal, retinoylβ-glucuronide 6′,3′-lactone (RBGUL), doconexent, 4,8,13-duvatriene-1,3-diol (DTD) and 2,4-di-tertbutylphenol (2,4-DtBP) demonstrated improved target binding affinity. Amongst these, retinal, RBGUL and 2,4-DtBP were selected as potential antiviral candidates that effectually blocked the viral spike protein. 2,4-DtBP binds the SARS-CoV-2 S1 RBD with -5.3 kcal/mol binding energy and associated with active site residues, Asn501 and Gly496 via hydrogen bonds as well as with Tyr505 and Arg403 through hydrophobic interactions. The fatty acid doconexent extracted from microalgae is rich in docosahexaenoic acid (DHA), which has enhanced anti-inflammatory characteristics and therefore has been repurposed to treat SARS-CoV-2 infection (Singhal et al., 2020; Stanly et al., 2020) [269,270]. The most absorbable vitamin A aldehyde, retinal has been used in treating SARS-CoV-2 infection (Michele et al., 2020; Morais et al., 2020; Gröber and Holick 2021) [271,272,273]. Kulkarni et al., 2021 [268] also showed that RBGUL has an elevated binding affinity of −7.0 kcal/mol to SARS-CoV-2. In this context, in silico investigations play an important role in the process of early antiviral drug discovery. 

Nallusamy et al., 2021 [274], examined the inhibition of multiple protein targets of SARS-CoV-2 through virtual screening, wherein they studied 605 phytochemicals obtained from 37 plant species and 139 antiviral compounds (Pubchem and Drug bank). Results showed that SARS-CoV-2 MPro differed significantly rom MERS-CoV MPro and SARS-CoV MPro, therefore necessitating the discovery of novel drugs. When the phytochemical cyanin obtained from Zingiber officinale was screened for inhibitory effects against major proteases of all three major Coronaviruses, it showed binding energies of −7.7 kcal/mol, −8.2 kcal/mol and −8.3 kcal/mol against these proteases of MERS-CoV, SARS-CoV and SARS-CoV-2, respectively. Agathisflavone, amentoflavone, chlorogenin and catechin-7-o-gallate exhibited inhibitory activities against multiple targets. *Anacardium occidentale*, *Azadirachta indica*, *Clerodendrum serratum*, *Cissus quadrangularis*, *Mangifera indica*, *Pedalium murex*, *Solanum nigrum*, *Terminalia chebula*, *Ocimum basilicum* and *Vitex negundo* have also been shown as potential sources of anti-SARS-CoV-2 phytochemicals. Intriguingly, this investigation showed the importance and antiviral properties of the traditional Indian herbal formulation called “Kabasura kudineer”, endorsed by the AYUSH unit of the Indian Government. 

Sourced from 22 plants, 33 phytochemicals were shown to be the best hits with increased binding affinities towards SARS-CoV-2 targets. Five compounds, agnuside, luteolin 7-O-beta-D-glucoside, friedelin, luteolin-7-o-betad-glucopyranoside and luteolin 7-O-(6′-malonylglucoside), exhibited high binding affinities towards SARS-CoV-2 protein targets such as the major protease, spike glycoprotein, NSP3, NSP9 and NSP15. *Solanum nigrum* has been reported to contain phytochemicals such as Solasodine, Solanocapsine, Diosgenin and Spirostan-3-ol, N-methylsolasodine, all of which effectively blocked three targets in the SARS-CoV-2, i.e., NSP9, NSP16-NSP10 and MPro.

*Pedalium murex* is the source of phytochemicals such as Lupeol acetate (targeting NSP16-NSP10), Rubusic acid (targeting SARS-CoV-2 Mpro, NSP3), Diosgenin (targeting NSP9) and Urosolic acid (targeting NSP15) which could be considered as candidates for SARS-CoV-2 inhibition. Medicinal plants such as *Ocimum basilicum*, *Cissus quadrangularis*, *Azadirachta indica*, *Clerodendrum serratum* and *Terminalia chebula* are reportedly sources of phytochemicals capable of inhibiting SARS-CoV-2 targets. Such herbal plants could be the focus of future research in alleviating SARS-CoV-2 and other emerging coronavirus infections. Comprehending the composition of phytochemicals in these plants will enable the discernment of the functions of different bioactive molecules used in traditional medical practices such as Siddha and Ayurveda (Nallusamy et al., 2021) [274]. Cyanin sourced from Zingiber officinale was identified to be the best phytochemical having the highest binding energy towards major proteases of all three Coronaviruses. The cashew nut (*Anacardium occidentale*) and mango (*Mangifera indica*) have been found to be rich in agathisflavone and amentoflavone which exhibited potential inhibitory activity against multiple molecular targets of the SARS-CoV-2. Phytochemicals from *Carica papaya* have been shown to exhibit inhibitory activities against the MPro and spike glycoprotein of SARS-CoV-2. 

Computational studies recently conducted (Balkrishna et al., 2021) [275] showed that withaferin A and withanone, phytochemicals sourced from *Withania somnifera,* targeted the SARS-CoV-2 and could be considered as viral entry inhibitors. In silico investigations of the biochemical interactions between the SARS-CoV-2 ACE2-RBD and withanone were conducted using molecular dynamics simulation, molecular docking and calculation of electrostatic energy followed by subsequent biochemical validation. Additionally, extracts from *W. somnifera* enriched in withanone were tested for their capability to alleviate SARS-CoV-2 associated pathological features in humanized zebrafish through the induction of the recombinant SARS-CoV-2 S protein. Withanone was found to bind the ACE2-RBD complex at the interface of their interaction, resulting in energetic destabilization. The electrostatic aspect of the binding free energies of this complex was also notably decreased. The interchain long-range ion-pair (K31-E484) and the two intrachain salt bridge interactions (K31-E35) at the interface of the ACE2-RBD were totally eliminated by withanone in the 50 ns simulation (Balkrishna et al., 2021) [275]. Withanone effectually repressed the ACE2-RBD interaction with an IC50 value of 0.33 ng/mL in a dose-dependent manner, as demonstrated by in vitro binding assays. Therefore, this study provided biochemical and in vivo validation of the computational findings on the potential of withanone as a strong inhibitor of SARS-CoV-2 entry into host cells. 

Kato et al., 2021 [276] reported on food phytochemicals including theaflavin, epigallocatechin gallate, myricetin, piceatannol, herbacetin, isothiocyanates and myricitrin inhibited SARS-CoV-2 MPro enzyme with IC50 values ranging from 0.4 to 33.3 μM against the 0.5- μM of the enzyme. Piceatannol is a resveratrol metabolite abundantly produced in Passiflora edulis fruits (Matsui et al., 2010) [277]. Piceatannol inhibits the SARS-CoV-2 MPro enzyme wherein its catechol structure was found to be critical for the interaction with the enzyme. The covalent interaction of myricetin and epigallocatechin gallate to the cysteine residue located in the active site of MPro was demonstrated. In previous studies, some flavonoids such as quercetin 3-O-glucoside and herbacetin were shown to inhibit MPro of MERS-CoV (Jo et al.2019]) [278], while flavones and flavonoids such as pectolinarin, rhoifolin, apigenin, herbacetin, quercetin, amentoflavone and luteolin were shown to inhibit MPro of SARS-CoV (Jo et al., 2020; Ryu et al., 2010) [108,279]. On the other hand, phytochemicals such as theaflavin (Jang et al., 2020), quercetin [Abian et al., 2020], epigallocatechin gallate (EGCG) Jang et al., 2020; Chiou et al., 2021, myricetin (Su et al., 2021), pectolinarin, herbacetin, baicalin (Jo et al., 2020), baicalein, rutin (Deetanya et al., 2021), as well as cyanidin-3-glucoside (Pitsillou et al., 2020) inhibited SARS-CoV-2 MPro in vitro [280,281,282,283,284,285,286,287,288,289]. 

Mahmud et al., 2021 [287] conducted both in silico and in vitro analyses to identify important *A. officinalis* phytochemicals capable of inhibiting the MPro active sites of SARS-CoV-2. They studied the antioxidant activities of *Avicennia officinalis* fruit and leaf extracts out of which four potent substances, methyl linoleate, tricontane, methy palminoleate and hexacosane complexed with the MPro of SARS-CoV-2 with binding free energy values of −6.3, −6.75, −6.3 and −6.7 kcal/mol, respectively. The active site amino acid residues, Arg188, Cys145, Gln 189, Glu166 and Met165 in the MPro were shown to form non-bonded interactions with these compounds. 

Plant-derived polyphenols such as apigenin (Ryu et al., 2010a) [108], myricetin (Yu et al., 2012) [110], kaempferol (Schwarz et al., 2014) [288], resveratrol (Wahedi et al., 2020) [289] and quercetin (Chiow et al., 2016) [290] have been shown to exhibit significant activities against CoVs. Geranylated flavonoids (tomentin A-E) sourced from *Paulownia tomentosa* (Thunb.) Steud. (*Paulowniaceae*) were shown to inhibit the SARS-CoV papain-like protease (Cho et al., 2013) [291]. Additionally, flavonoids including herbacetin, pectolinarin and 7-O-rhamnoglucoside at 20 µM concentration inhibited the critical enzyme for the replication of SARS-CoV, 3C-like protease (Jo et al., 2020 [279]). This enzyme was also repressed with 10 polyphenols sourced from *Broussonetia papyrifera* (L.) L’Hér. ex Vent. (*Moraceae*), particularly with papyriflavonol A at 3.7 µM (Park et al., 2017) [292]. 

Molecular docking studies on anti-SARS-CoV-2 compounds used in traditional Chinese medicine revealed that the flavonoid theaflavin obtained from black tea, *Camellia sinensis* (L.) Kuntze (Theaceae), could inhibit the RNA-dependent RNA polymerase of SARS-CoV-2 (Lung et al., 2020) [293]. Further, hesperidin, found abundantly in citrus, showed potential for ACE2 inhibition and can be considered as an effectual candidate for SARS-CoV-2 clinical trials (Haggag et al., 2020) [294]. Alkaloids have also been shown to have anti-CoV effects. The indolizidine alkaloid lycorine sourced from *Lycoris radiata* (L’Hér.) Herb. (*Amaryllidaceae*) effected anti-SARS-CoV activities at a concentration of 15.7 nM (Li et al., 2005) [104]. Docking studies showed that an indole alkaloid, 10-hydroxyusambarensine sourced from *Strychnos usambarensis* Gilg ex Engl. (Loganiaceae), and a bisnorterpene, 6-oxoisoiguesterin obtained from *Salacia madagascariensis (Celastraceae*) exhibited anti-SARS-CoV-2 effects by binding to the virus 3C-like protease (Gyebi et al., 2020) [295]. Bisbenzyliso-quinoline alkaloids, fangchinoline (1.01 µM), cepharanthine (0.83 µM) and tetrandrine (0.33 µM) isolated from *Stephania tetrandra* S. Moore (*Menispermaceae)* are other natural compounds showing anti-HCoV activities (Kim et al., 2019) [296]. Other alkaloid compounds such as berberine, emetine, chelidonine, jatrorrhizine, palmatine, ipecac alkaloids and sanguinarine have been suggested as antiviral agents against SARS-CoV-2 (Bleasel and Peterson 2020; Wink 2020) [297,298]. Triterpene saponin glycosides and saikosaponins (A, B2, C, D) sourced from *Bupleurum* spp. (*Apiaceae)* have been shown to have anti-CoV effects. At 6 μM, the saikosaponin B2, showed anti-CoV effects in addition to inhibiting virus propagation (Cheng et al., 2006) [92]. Another triterpene saponin, glycyrrhizin sourced from *Glycyrrhiza glabra* L. (*Fabaceae*), showed anti-CoV activity in addition to blocking multiple stages of viral replication such as adsorption and permeation (Bailly and Vergoten 2020; Cinatl et al., 2003) [99,100]. Triterpenoids isolated from *Euphorbia neriifolia* L. (Euphorbiaceae) also demonstrated anti-HCoV effects, amongst which friedelane derivatives showed the highest effect (Chang et al., 2012) [299]. Phytochemicals contained in essential oils are some other natural compounds with anti-HCoV effects (Nadjib 2020) [300]. Monoterpenes, 1,8-cineole and jensenone isolated from the essential oils of *Eucalyptus* spp. (*Myrtaceae*) displayed anti-CoV effects in molecular docking studies (Sharma and Kaur 2020a; Sharma and Kaur, 2020b) [301,302]. The lignoid, savinin obtained from *Chamaecyparis obtusa* (Siebold & Zucc.) Endl. (*Cupressaceae*) and the triterpenoid, betulinic acid obtained from *Betula* spp. (*Betulaceae*), proved to have anti-SARS-CoV activities through 3CL protease inhibition at 25 and 10 μM, respectively (Wen et al., 2007) [109]. Furthermore, quinones including aloe-emodin, emodin and quinones from *Tripteryguim regelii* (*Celastraceae*) such as iguesterin, tingenone, pristimererin and celastrol also exhibited anti-SARS-CoV activities. Iguesterin, emodin and aloe-emodin were found to inhibit the SARS-CoV 3CL protease at 2.6, 20 and 366 µM, respectively (Lin et al., 2005; Ryu et al., 2010b; Schwarz et al., 2011) [112,303,304]. Additionally, emodin sourced from both *Polygonum multiflorum* (Thunb.) Moldenke (*Polygonaceae*) and *Rheum officinale* Baill., blocked the S protein-ACE2 interaction at 200 µM. Taken together, it is evident that phytochemicals are propitious sources for the identification of effectual antiviral agents against CoVs, particularly, SARS-CoV-2. 

Many clinical trials on anti-CoV phytochemicals have been formulated and are currently being conducted (Majnooni et al., 2020) [305]. These include resveratrol (NCT04542993, NCT04536090, and NCT04377789), artemisinin and curcumin (NCT04382040), hesperidin and diosmin (NCT04452799), polyphenols (NCT04400890), epigallocatechin gallate (NCT04446065), quercetin (NCT04468139 and NCT04377789), colchicine (NCT04527562, NCT04392141, NCT04375202, NCT04355143, and NCT04360980), glycyrrhizin (NCT04487964), tetrandrine (NCT04308317) and berberine (NCT04479202). 

Wang et al., 2022 [306] report that investigating the relationship between SARS-CoV-2 infection and the p38 cellular protein could be of great importance in deciphering potential immune regulation and development of novel intervention measures against this virus. SARS-CoV infection induces MKK3/6 phosphorylation in Vero E6 cells and further augments p38 expression. SARS-CoV-2 infection elicits excessive p38 MAPK activation following viral entry (Grimes and Grimes 2020) [307]. p38 MAPK activation induces receptor-mediated endocytosis of SARS-CoV-2, thus facilitating entry of the virus (Zhou et al., 2020) [308]. Additionally, the PLpro enzyme encoded by SARS-CoV induces p38 activation. Grimes and Grimes 2020 [307] report that, upon knock down of MAP2K3, p38δ and p38γ by siRNA in ACE2-expressing A549 cells, SARS-CoV-2 replication was notably suppressed. Pharmacological inhibition of p38 activation could thus be a propitious SARS-CoV-2 intervention strategy and p38 MARK inhibitors could have significant potential against infection by SARS-CoV-2. Plant-derived compounds could be highly beneficial as p38 inhibitors and further studies are warranted in exploring this avenue. 

Even as vaccines have been generated to preclude infection and most of the Covid19 cases resulting in mild infections, there are many severe cases of the disease that end up in fatalities even with adequate treatment. Therefore, new strategies to confront the effects of this disease could prove to be essential in saving lives. Presently, there have been several clinical trials involving novel oral antivirals demonstrated to be effectual in Covid19 treatment, but only upon administration in the early days of SARS-CoV-2 infection before hospitalization (Pfizer 2021; Kabinger et al., 2021; Gumbrecht et al., 2021) [309,310,311]. Two of these antiviral oral medications have received emergency FDA approval (Gumbrecht et al., 2021) [311] including (1) Paxlovid from Pfizer which inhibits the SARS-CoV-2 chymotrypsin-like 3CL protease that plays an essential role in viral replication (Pfizer, 2021; Vandyck and Deval, 2021) [309,312] and (2) molnupiravir from Merck which is metabolized into a ribonucleoside analogue (β-D-N4 -hydroxy-cytidine triphosphate) in the human body and this analogue is incorporated into the SARS-CoV-2 RNA leading to lethal mutagenesis in the process of viral replication. These drugs are more efficacious compared to other similar antivirals such as the intravenously administered remdesivir (Kabinger et al., 2021; Gumbrecht et al., 2021) [310,311]. However, despite the development of these new effective treatments, there is still a compelling need to discern the SARS-CoV-2 molecular machinery to identify new targets for therapy as the above antivirals work only upon administration at the early stages of SARS-CoV-2 infection. Deciphering the molecular mechanisms associated with rare severe side effects occurring in response to the currently used Covid19 vaccines could result in better measures for monitoring and management of risks of developing such side effects. 

Adamalysin (ADAM) proteins have diverse roles in the regulation of various functions in humans besides having important roles in inflammation (Dias et al., 2022) [313]. These proteins, in particular, ADAM17, play an important role in ACE2 receptor shedding, while enhancing the entry of SARS-CoV-2, promoting inflammation and disease pathogenesis (Haga et al., 2008, 2010) [314,315]. Novel therapies to combat severe forms of the SARS-CoV-2 are still required and given the role of SARS-CoV-2 in many cellular processes such as cell signaling, fibrosis or inflammation [Theret et al., 2021; Edwards et al., 2008] [316,317], the adamalysin proteins could prove as likely new targets for therapy of Covid19 disease. Trafficking of phosphatidylserine, an acidic, highly negatively charged phospholipid in the membrane of mammalian cells to the outer side provides a docking site for protein kinase C, an enzyme implicated in the induction of ADAM17 activity. This exposure of the phosphatidylserine is essential for the sheddase activity of ADAM17 induced by a conformational alteration in its structure that pulls the catalytic domain of this enzyme closer to the surface of the cell to facilitate the cleavage of substrates. SARS-CoV-2 has been shown to mediate the exposure of phosphatidylserine, which may be a mechanism for cleavage of ACE2, IL-6R, TNF-α, and other pro-inflammatory factors that are important components of inflammatory processes induced by SARS-CoV-2 (Arganaraz et al., 2020; Lambrecht et al., 2018) [318,319]. Plant-based phytochemicals could be of great applicability as inhibitors of adamalysins. Hence, investigating the roles of adamalysins in Covid19 disease may provide clues into the molecular mechanisms of SARS-CoV-2 in addition to exploring novel therapeutic targets. 

## 15. Regulatory and Safety Aspects of the Application of Phytomedicines and Plant-Based Vaccines/Therapeutics

The search for effectual phytochemicals against SARS-CoV-2 still continues and, considering the novelty of the Covid19 virus and the resulting disease, safety concerns loom large over the application of phytomedicines [88]. In this regard, there is a dearth of available data on the safety, efficacy and quality of the majority of these plants. Uninvestigated use of purified compounds is counter-recommended and these phytochemicals must first undergo proper testing for safety and efficacy. However, despite the fact that every plant product must be properly investigated for unwanted side effects, phytochemicals are inherently safe to use by virtue of their biocompatibility, eco-friendliness and diminished toxicity when compared to xenobiotics. 

The FDA has provided directives for the safe use of natural compounds as drugs [320]. Likewise, the European Medicines Agency (EMA) has mandated that herbal medicinal compounds can only be used if they have been under usage for at least 3 decades, which includes at least 15 years in the European Union, and are not administered parenterally. Moreover, marketing, and well-established application of the products has to be authorized only after there is availability of sufficient scientific data confirming that the active ingredient or pure constituent of the herbal product has recognized potency [321,322]. Nevertheless, there is a mistaken perception that herbal products are completely safe and devoid of any adverse side effects. There are numerous toxic compounds in plants. In comparison with synthetic compounds, plant-based compounds have demonstrated lowered toxicity and therefore require less rigorous evaluation. Notwithstanding, to preclude the side effects caused by the use of these plant-derived natural products in an unregulated manner, safety regulation on their use is important [323]. Only then can promising plant-based drug candidates against other Coronaviruses be considered for their effectual use against Covid19. 

Researchers world-wide are involved in serious efforts to utilize all of the available platforms for the development of safe, effective vaccines against Covid19. With the inception of transient expression technology, plant-based production could be deemed as a viable strategy that is recently gaining the interest of various pharmaceutical companies. Recombinant biopharmaceuticals such as therapeutic proteins and vaccine antigens can be produced in plants at large scales within brief time periods. Several research teams within the plant molecular farming community have ventured into the development of plant-based vaccines. Nevertheless, the use of plant-based vaccines is subject to the regulatory pathway, which is often time-consuming and complex. The benefits of plant-based expression systems can only be realized by surmounting regulatory hurdles. 

It is essential to harmonize regulatory procedures for plant-based products both at the international and national levels in order to reduce the time span of plant biologics to be translated from the bench level to the market. Moreover, these plant-derived products must attain good quality standards and meet with the guidelines of stringent GMP regulations assigned for biological products. Presently, very few plant-generated vaccines against SARS-CoV-2 have reached up to the stages of preclinical and clinical trials. Despite the slow progress in the commercialization of plant-based vaccines, the success of clinical trials for plant-derived vaccines against Covid19 and influenza in recent years provides great promise towards the commercialization of such plant-made vaccines in the near future. 

## 16. Conclusions and Future Perspectives

Within the last 2 decades, three extremely pathogenic Coronaviruses, MERS-CoV, SARS-CoV and SARS-CoV-2, have been identified in the human population. Considerable advances have been made in developing plant-based vaccines and phytochemicals against these viruses through the application of current knowledge and proof-of-concept studies on other vaccine/therapeutic candidates derived from plants. Promising outcomes have been reported regarding plant-generated SARS-CoV-2 vaccine candidates. Similar strategies based on subunit vaccines can be examined for efficacy and safety studies. Particularly, glycosylation plays an important role in determining the antigenic properties of the respective vaccine antigens. Despite the fact that plants are capable of performing post-translational modifications similar to mammalian and human cells, any discrepancies in the glycosylation patterns of the plant-derived mAbs can be circumvented in plants by glycoengineering to obtain humanized forms of the respective proteins [324] that could be used to generate SARS-CoV-2 vaccine candidates in plants. Plant-based vaccines serve as cost-efficient platforms to produce Covid19 vaccines which enhances their feasibility to be employed in immunization programs on large scales. The evaluation of SARS-CoV-2 subunit and VLP vaccines raised in plants could contribute substantially in advancing this field. Nevertheless, plant-based vaccines need to be developed further to obtain broad-spectrum immuno-protection against re-emerging variants of the virus. 

Despite the fact that there exist similar features in the diseases caused by the three major Coronaviruses, distinctive infection characteristics and features have been elucidated in the clinical outcomes and the immunopathology of each of these viral infections. Frequent outbreaks of pathogenic and highly infectious strains of Coronaviruses have caused great burden and threat to the population such as the ongoing Covid19 pandemic that has led to unforeseen health crisis having devastating health and socio-economic outcomes worldwide. All of the three CoVs share immunological features affecting pathological attributes. These viruses replicate in the immune cells of the host which triggers innate immune reactions leading to the elicitation of cytokines and pro-inflammatory cells. Such cytokine storm leads to life-threatening outcomes and, ultimately, the body reacts by generating protective antibodies which clear the infection while conferring immunity to further infection by the same virus. 

In vitro investigations have been greatly helpful in discerning the pathological and immunological features of the viruses as well as in performing drug trials to combat these agents. Nevertheless, there is a compelling need for the advancement of clinical research studies, as these viruses can mutate further leading to the generation of variants with increased pathogenicity. Intense, collaborative measures by scientists the world over have enabled advanced discoveries concerned with several aspects of infections against Coronaviruses. Moreover, characterizing the immunopathological and clinical facets of their infection will be of tremendous help in designing safer and more efficacious drugs, vaccines and medicaments to combat the emergence and re-emergence of these pathogenic Coronaviruses. Potential outcomes of clinical investigations of various plant-based antiviral drug candidates and vaccines afford hope that the current pandemic will end soon, and will further help in stemming the tide of any future pandemics. 

## Figures and Tables

**Figure 1 vaccines-10-01805-f001:**
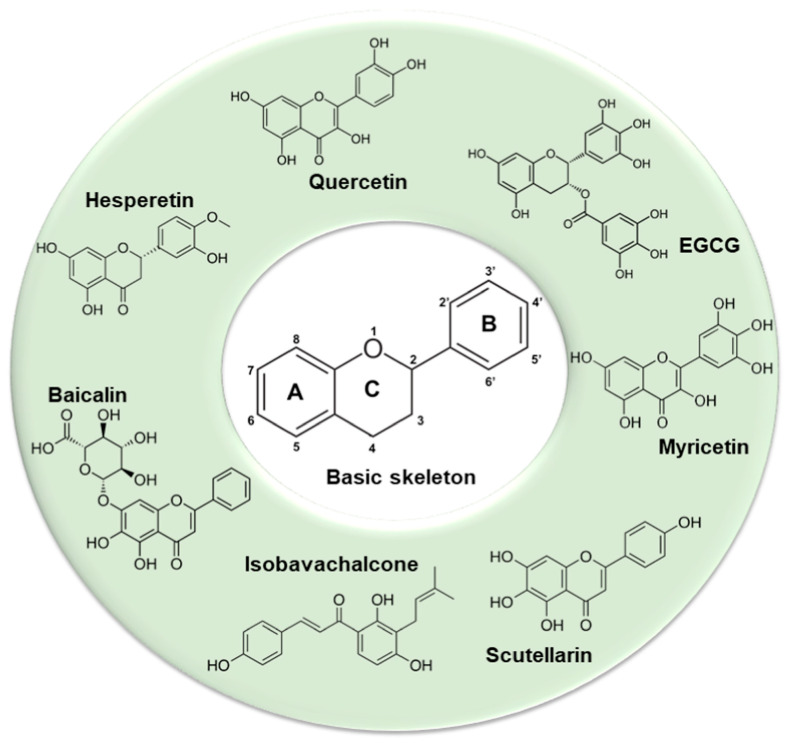
Basic skeleton (C6–C3–C6) of flavonoids and representative examples of compounds able to counteract coronavirus infection. (Reproduced from [111]).

**Figure 2 vaccines-10-01805-f002:**
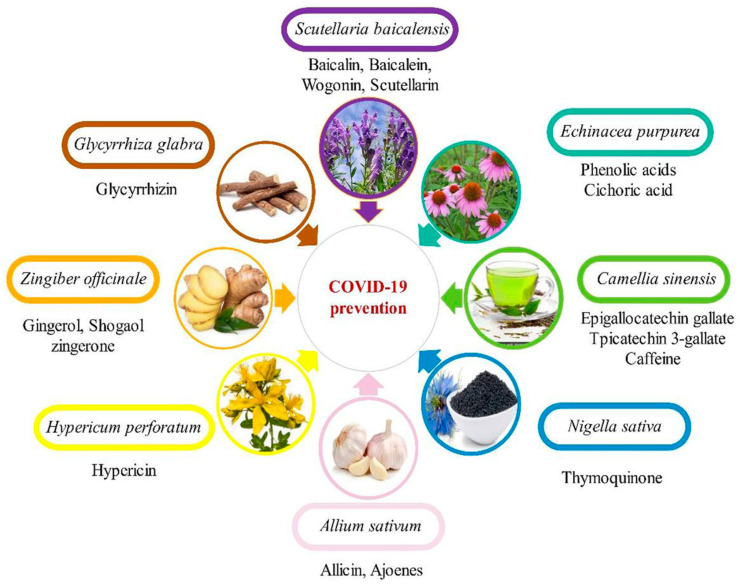
Some natural plant products for precluding SARS-CoV-2 infection (Reproduced from [122]).

**Table 1 vaccines-10-01805-t001:** Important features of Coronaviruses (Adapted from [1]).

**MERS** Family: *Coronaviridae* Genus: *Betacoronavirus* Lineage: C				
**Genome & Structural Proteins**	**Receptor**	**Organs Expressing the Receptor**	**Transmission & Epidemiology**	**Symptoms, Diagnosis & Vaccines**
30.1 kb ssRNA (+); Envelope protein (E), Spike protein (S), Membrane protein (M), Nucleocapsid protein (N)	Spike protein RBD interacts with the receptor, Human dipeptidyl peptidase 4 (DPP4 or CD26)	Lung, liver, heart, brain, spleen, kidney and intestine	Disease: MERS reported first in Saudi Arabia, 2012 Natural host: bats Intermittent host: dromedary camels Mode of transmission: human-to-human Currently active, not pandemic No viral variants Fatality rate: ~35%	Latency period: 5–12 days after viral exposure; Flu-like symptoms, gastrointestinal complications, acute respiratory distress syndrome (ARDS), septic shock, multi-organ failures and respiratory failure; Diagnosed by RT-PCR/antibody tests 3 vaccines in clinical phase
**SARS-CoV** Family: *Coronaviridae* Genus: *Betacoronavirus* Lineage: B				
29.7 kb ssRNA (+) Envelope protein (E), Spike protein (S), Membrane protein (M), Nucleocapsid protein (N)	Spike protein RBD interacts with the receptor, Human angiotensin converting enzyme 2 (ACE2)	Lung, intestine, kidneys, heart, liver and testicles	Disease: SARS; Reported first in China, 2003 Natural host: bats Intermittent host: Palm civet cats Mode of transmission: human-to-human Currently disappeared, not pandemic No viral variants Fatality rate: ~10%	Latency period: 2–7 days after viral exposure but may be as long as 10 days; Flu-like symptoms, progressing to hypoxemia, severe respiratory illness, low white blood cell counts and low platelet counts; Diagnosed by RT-PCR/antibody tests 2 vaccines in clinical phase
**SARS-CoV-2** Family: *Coronaviridae* Genus: *Betacoronavirus* Lineage: B				
29.9 kb ssRNA (+) Envelope protein (E), Spike protein (S), Membrane protein (M), Nucleocapsid protein (N)	Spike protein RBD interacts with the receptor, Human angiotensin converting enzyme 2 (ACE2)	Lung, intestine, kidneys, heart, liver and testicles	Disease: Covid19 Reported first in China, 2019 Natural host: bats Intermittent host: Pangolins Mode of transmission: human-to-human Currently ongoing pandemic Viral variants: Alpha, Beta, Gamma, Delta, Lota, Kappa, Eta, Lambda, Omicron Fatality rate: ~2.2%	Latency period: 2–12 days after viral exposure; Flu-like symptoms, chills, coughing, fatigue, breathlessness, muscle ache, headache, sore throat, congestion, loss of smell or taste; Gastrointestinal symptoms: nausea, vomiting, diarrhea Severe complications: pneumonia, acute respiratory syndrome, liver injury, myocarditis, acute kidney injury, neurological complications, cardiopulmonary failure, acute cerebrovascular disease, shock; Diagnosed by RT-PCR and other laboratory techniques; 47 vaccines approved and 89 vaccine candidates in clinical trials

N/A: Not available.

**Table 2 vaccines-10-01805-t002:** List of plant-based vaccine candidates against Coronaviruses.

**a. Studies at the Research Phase Using Transient Expression in *N. benthamiana*.:**			
**Vaccine Candidate Antigen**	**Vaccine Composition, Route of Administration and Formulation**	**Immunization Potential**	**Reference**
SARS-CoV S1-GFP fusion protein	N/A	N/A	[165]
SARS-CoV recombinant N and M proteins	N/A	N/A	[166]
SARS-CoV-2 recombinant N protein & recombinant RBD protein	N/A	N/A	[167]
**b. Studies at the Pre-Clinical Phase Using Transient Expression in *N. benthamiana*.:**			
**Vaccine Candidate Antigen**	**Vaccine Composition, Route of Administration and Formulation**	**Immunization Potential**	**Reference**
SARS-CoV nucleocapsid protein	Formulated with Freund’s adjuvant and IP with at 2-week intervals for four doses	Able to induce humoral immunity as well as SARS-CoV-2 cytokine-producing cells in mice	[163]
SARS-CoV-2 VLP-based IBIO-200 vaccine generated by iBio, Inc. (Bryan, TX, USA)	IM injection on day 1 and 21	Able to stimulate specific immune responses and neutralizing antibody against SARS-CoV-2 in mice	[168]
SARS-CoV-2 IBIO-201 spike-based sub-unit vaccine generated by iBio, Inc. (Bryan, TX, USA)	Formulated with LicKMTM adjuvant and IM injection on day 1 and 21	Able to stimulate specific immune responses and neutralizing activities against SARS-CoV-2 in mice better than IBIO-200	[168]
SARS-CoV-2 IBIO-202 Nucleocapsid protein-based sub-unit vaccine produced by iBio, Inc. (Bryan, TX, USA)	N/A	Able to induce robust, antigen-specific, memory T cell response	[169]
Baiya Vax 1 subunit vaccine generated by Baiya Phytopharm Co., Ltd. (Bangkok, Thailand)	Formulated with alum adjuvant and IM injection on day 1 and 21	Able to induce antigen-specific IgG and neutralizing responses as well as cellular immunity in mice and non-human primates	[170]
**c. Studies at the Stage of Clinical Trials/Already Approved.**			
**Vaccine Candidate Antigen**	**Vaccine Composition, Route of Administration and Formulation**	**Immunization Potential**	**Reference**
SARS-CoV-2 KBP-201 RBD-based vaccine developed by British American Tobacco (BAT) and its subsidiary Kentucky BioProcessing, Inc. (Owensboro, KY, USA)	Formulated with CpG adjuvant and IM injection on day 1 and 22	Able to induce positive SARS-CoV-2-specific immunity in pre-clinical trials	Phase I/II ClinicalTrials.gov Identifier: NCT04473690 [171]
SARS-CoV-2 VLP-based Covifenz vaccine with AS03 adjuvant developed by Medicago Inc. (Quebec City, QC, Canada) and GlaxoSmithKline (GSK)	IM injection given 21 days apart	Able to induce antibody responses in the volunteers with two doses of immunization; 75.3% efficacy against the Delta variant; booster dose for evolving variants such as Omicron being currently engineered	Approved in Canada Phase 2/3 in Argentina, Brazil, Japan, Canada, United Kingdom of Great Britain and Northern Ireland, United States of America NCT04636697 [trackvaccines.org] [172] Only plant-based vaccine to be approved

N/A: not available; GI: gastric intubation; IM: intramuscular immunization. (Adapted from [1]).

**Table 3 vaccines-10-01805-t003:** (**a**) List of plant-produced anti-coronavirus mAbs generated using transient expression in *N. benthamaina* and currently at the research stage. (**b**) List of other therapeutic proteins and diagnostic reagents.

**(a)**		
**mAb**	**Essential Aspects**	**Reference**
Anti-SARS-CoV CR3022 mAb	Specific binding to RBD domain on S1 subunit of SARS-CoV and SARS-CoV-2	[145]
Anti-SARS-CoV nucleocapsid (N) CR3009-scFv	Specific binding to N protein of SARS-CoV and SARS-CoV-2	[167]
Anti-SARS-CoV nucleocapsid (N) CR3018-scFv		
Anti-SARS-CoV/CoV-2 nanobody72 VHH-Fc IgG1	Specific binding to RBD domain on S1 subunit of SARS-CoV-2	[167]
Anti-SARS-CoV-2 B38 and H4 mAb	Specific binding to RBD domain on S1 subunit of SARS-CoV-2 and exhibited neutralizing activity against viral infection in vitro	[205]
Anti-SARS-CoV-2 sybody3 VHH-Fc IgG1	Specific binding to RBD domain on S1 subunit of SARS-CoV-2	[167]
Anti-SARS-CoV-2 sybody17 VHH-Fc IgG1		
**(b)**		
**Proteins against Coronaviruses Using Transient Expression in *N. Benthamiana* and Currently at the Research Stage.**		
**Formulation (Kit/Protein)**	**Essential Aspects**	**Reference**
ACE2-Fc fusion protein developed by iBio, Inc. (Bryan, TX, USA)	ACE2-Fc precludes SARS-CoV-2 virus from infecting Vero E6 cells	[217]
ACE2-Fc fusion protein developed by Baiya Phytopharm Co., Ltd. (Bangkok, Thailand)	Blocking and neutralizing RBD domain of the SARS-CoV-2 S1 subunit	[147]
**Other Diagnostic Reagents Currently at the Production Phase**		
Recombinant SARS-CoV-2 nucleocapsid protein generated by Leaf Expression System (Norwich, UK)	Used as an antigen diagnostic reagent	[218]
Baiya rapid SARS-CoV-2 IgM/IgG test kit developed by Baiya Phytopharm Co., Ltd. (Bangkok, Thailand)	Used as lateral-flow immunoassay strip reagents for detecting IgM/IgG antibodies in human sera	[216]
SARS-CoV-2 RBD-based ELISA test kit developed by Diamante, Italian biotech company	Used in ELISA for the detection of serum antibody levels in SARS-CoV-2 convalescent patients	[141]

Adapted from [1].

## Data Availability

Not applicable.

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
