# Peer review of "Plant Molecular Pharming and Plant-Derived Compounds towards Generation of Vaccines and Therapeutics against Coronaviruses"

_vaccines, 2022, doi:10.3390/vaccines10111805_

Round 1

Reviewer 1 Report

The study is quite detailed and interesting. However whats new. Is it a summary of previous studies. I don't see any new information coming. Thats my major concern before publication.

Author Response

Answer: Newly published material has been included under the section ‘Recent developments regarding intervention strategies against coronaviruses’. 

Reviewer 2 Report

The Review of Venkataraman S. describes recent developments in using plant-derived extracts and their related compounds for the generation of novel vaccines and therapeutic proteins against Coronaviruses. The manuscript is clear and discusses well previously published data; however, Review should be edited by a native speaker for English language and grammar (punctuation).

Tables 1-3 should be modified and updated, because presently they are similar to Tables 1, 3, and 4 from the Review of Shanmugaraj et al. Pathogens 2021;10(8):1051; doi: 10.3390/pathogens10081051. According to the Vaccines – COVID19 Vaccine Tracker (trackvaccines.org) website there are 225 vaccine candidates in clinical trials, including 87 vaccines in phase 3 (update 02 September 2022), Table 1 should be updated.

Author Response

Typos and punctuations fixed. 

All 3 tables, Table 1, 2 and 3 have been modified compared to that of the Shanmugaraj et al., Pathogens 2021;10(8):1051; doi: 10.3390/pathogens10081051. In Table 2, the Medicago generated Covifenz vaccine status has been updated to ‘approved’ (kindly observe the last part of the table). Also, the information that ‘to-date there are 227 vaccine candidates in clinical trials, including 47 vaccines already approved and 89 vaccines in phase 3 clinical trials’ has been mentioned in the text just after Table 2. From the trackvaccines.org website, I could identify only the Medicago and the Kentucky Bioprocessing vaccines as plant-based vaccines and therefore only this information is entered into Table 2 in accordance with the theme of this manuscript.

Reviewer 3 Report

The manuscript reviews the current knowledge of the molecular virology, infection process and disease progression of major Corona Viruses, focusing on MERS-CoV, SARS-CoV and SARS-CoV-2, and antiviral targets present in these viruses. The recent developments in the generation of plant-based antivirals and vaccines are also discussed as potential significant contributors against current and future CoV infections.

The authors advocate the use of plant-derived therapeutics due to them allowing the circumvention of several challenges in production of biopharmaceuticals as they function in a rapid, efficacious manner for large scale manufacturing in bulk, and are capable of generating therapeutic proteins and other biologics that mitigate disease, often with fewer side effects compared to allopathic medicines.

This is an in-depth look into the viruses responsible for significant CoV infections in the last few years as well as into therapeutic targets and development and production of plant-based therapies, or molecular pharming as the authors put it, for these and future CoV epidemics, offering a significant but summarized report on a pathway for CoV therapies. I would recommend this manuscript for publication after revision.

As a note, in pages 8, 14-16 and 18-22 there are several species mentioned but their scientific names are not italicized, as per convention they should be. In addition, it seems that the recent info on latest development of drug target for CoV should also be included in this manuscript. For examples: 

https://doi.org/10.1016/j.biopha.2022.112970

https://doi.org/10.1017/erm.2021.29

Author Response

Answer: Scientific names of species have been italicized.

Answer: The essence of both papers as referred to by the reviewer in the above links have been addressed in the last part of the section ‘Recent developments regarding intervention strategies against coronaviruses’. 

Round 2

Reviewer 1 Report

Accept

Author Response

Thank you. 

Reviewer 2 Report

The author has responded well to the criticism. However, a few issues should be taken into consideration.

1. The record of percentage and binding energies [- 1.0 kcal.mol-1 or (-) 1.0 kcal/mol) should be unified in the manuscript.

2. There is a mess in the references. On pages 25 and 27 there are two references without the numbers (Kim et al., 2022 and Jassbi et al., 2017, respectively). Moreover, on the Reference list, there are 341 positions, while in the manuscript 336. The double enumerating on the references should also be avoided (round and square brackets).

Author Response

1. Record of percentage and binding energies is unified as per the reviewer's suggestion.

2. The list of references and their mention in the text have been duly corrected and double enumeration of references removed. 

Reviewer 3 Report

The authors have revised the manuscript accordingly. Thanks!

Author Response

Thank you.